# Lung Models to Evaluate Silver Nanoparticles’ Toxicity and Their Impact on Human Health

**DOI:** 10.3390/nano12132316

**Published:** 2022-07-05

**Authors:** Jesús Gabriel González-Vega, Juan Carlos García-Ramos, Rocio Alejandra Chavez-Santoscoy, Javier Emmanuel Castillo-Quiñones, María Evarista Arellano-Garcia, Yanis Toledano-Magaña

**Affiliations:** 1Programa de Maestría y Doctorado en Ciencias e Ingeniería (MyDCI), Facultad de Ciencias, Universidad Autónoma de Baja California, Ensenada 22860, Baja California, Mexico; jesus.gabriel.gonzalez.vega@uabc.edu.mx; 2Escuela de Ciencias de la Salud, Universidad Autónoma de Baja California, Blvd. Zertuche y Blvd., De los Lagos S/N Fracc, Valle Dorado, Ensenada 22890, Baja California, Mexico; juan.carlos.garcia.ramos@uabc.edu.mx; 3Tecnologico de Monterrey, Escuela de Ingenieria y Ciencias, Av. Eugenio Garza Sada 2501 Sur, Monterrey 64849, Nuevo Leon, Mexico; chavez.santoscoy@tec.mx; 4Facultad de Ciencias Químicas e Ingeniería, Universidad Autónoma de Baja California, Calzada Universidad 14418, Parque Industrial Internacional Tijuana, Tijuana 22390, Baja California, Mexico; castillo@uabc.edu.mx; 5Facultad de Ciencias, Universidad Autónoma de Baja California, Ensenada 22860, Baja California, Mexico; evarista.arellano@uabc.edu.mx

**Keywords:** silver nanoparticles, lung toxicity, cell lines, primary cultures, in vivo, in vitro, ex vivo monocultures, co-cultures, 3D cultures

## Abstract

Nanomaterials (NMs) solve specific problems with remarkable results in several industrial and scientific areas. Among NMs, silver nanoparticles (AgNPs) have been extensively employed as drug carriers, medical diagnostics, energy harvesting devices, sensors, lubricants, and bioremediation. Notably, they have shown excellent antimicrobial, anticancer, and antiviral properties in the biomedical field. The literature analysis shows a selective cytotoxic effect on cancer cells compared to healthy cells, making its potential application in cancer treatment evident, increasing the need to study the potential risk of their use to environmental and human health. A large battery of toxicity models, both in vitro and in vivo, have been established to predict the harmful effects of incorporating AgNPs in these numerous areas or those produced due to involuntary exposure. However, these models often report contradictory results due to their lack of standardization, generating controversy and slowing the advances in nanotoxicology research, fundamentally by generalizing the biological response produced by the AgNP formulations. This review summarizes the last ten years’ reports concerning AgNPs’ toxicity in cellular respiratory system models (e.g., mono-culture models, co-cultures, 3D cultures, ex vivo and in vivo). In turn, more complex cellular models represent in a better way the physical and chemical barriers of the body; however, results should be used carefully so as not to be misleading. The main objective of this work is to highlight current models with the highest physiological relevance, identifying the opportunity areas of lung nanotoxicology and contributing to the establishment and strengthening of specific regulations regarding health and the environment.

## 1. Introduction

Nanotechnology deals with matter manipulation at the nanoscale (1–100 nm) and has taken place in our lifetimes, becoming a worldwide trend. We can find it in various medical, cosmetic, and daily consumer products, most of which lack regulation [1]. Engineered nanomaterials (NMs) potentiate more promising applications than their bulk counterparts due to their unique physicochemical properties (PP). The most reported PP influencing NMs responses are size, shape, agglomeration, dissolution, surface charge, and surface reactivity [2]. Their small size and large surface/volume ratio endowed their high reactivity and the ability to surpass some biological barriers, both relevant for drug delivery and imaging applications [3].

Silver nanoparticles (AgNPs) are the most widely used nanomaterial today. These have found a wide range of applications in various areas such as textiles, agriculture, renewable energy, food, catalysis, bioremediation, and biomedicine [4,5,6]. Hence, their production will only increase in the upcoming years. Since the overproduction of AgNPs is inevitable, it is essential to figure out the possible environmental impact and the potential risk to human health. Inhalation, ingestion, and/or dermal contact are the main routes of exposure to nanoparticulate material, fibers, and silver vapors [7]. Numerous studies report the imminent dangers of inhaling nanoparticulate materials in work environments or in everyday life; harmful effects ranging from respiratory diseases to grayish discoloration of the skin in humans (argyria) and even possible DNA damage that could be expressed across generations [8,9]. Regardless of efforts to evaluate the toxicity of AgNPs in biological systems, their mechanism of action in the respiratory tract due to involuntary inhalation remains unclear to nanotoxicology. The lack of biosafety equipment to prevent the involuntary inhalation of NPs has opened a debate area to elucidate the possible consequences of short- and long-term exposure.

Many variables are involved in determining NMs’ cytotoxicity in the respiratory system. AgNPs can interact directly with the external or internal cellular membrane due to their morphological characteristics. Besides, biological factors such as biocorona formation or the nature of cells’ interactions with the AgNPs directly impact cell viability and proliferation [2]. AgNPs’ intrinsic characteristics such as concentration, coating nature, gender of the donor or animal model, and some extrinsic factors including exposure time and cell type were crucial in the cellular response exerted by AgNPs in vitro and in vivo [7,10]. Despite the advances in the in vitro culture models today, it has not been possible to perfectly simulate the cellular microenvironment of a respiratory system, specifically of the gas exchange interface, which serves as the primary standard when evaluating the harmfulness of AgNPs. Furthermore, different levels of model complexity exhibit different responses to silver interaction.

Even though silver has no biological role in the body and showed toxicity at specific doses to lower organisms, its use for medicinal purposes since ancient times is well documented [11]. However, how silver can act as a toxic agent against pathogenic bacteria, fungi, mold, parasites, viruses, cancer, and even in healthy human cells remained unclear. The toxicity exerted on model organisms has been mainly attributed to the release of Ag^+^ ions inside the cells, leading to reactive oxygen species (ROS) induction affecting various signaling pathways and mechanisms at transcriptional levels, known as transcriptional reprogramming [12,13].

Transcriptional reprogramming is often seen as a marker of disease, helping to elucidate harmful effects, and measured by genetic sequencing technics for a specific organ or tissue toxicity [14]. Monitoring punctual changes in the genome is widely known as the differential expression of genes (DEGs), a methodology used to identify and compile valuable datasets of over- and under-expressed genes in healthy and unhealthy cells associated with lung functionality and cell death pathways [15,16]. Hence, direct or indirect exposure to AgNPs’ toxicological profiles could be obtained from in vitro, ex vivo, and in vivo lung models and extrapolated to determine the impact of AgNP formulations on human health. In its different presentations, silver promises to be an excellent alternative to conventional lung chemotherapeutics, with advantages such as the incidence decrease of ventilator-associated pneumonia by coating tubes with varying formulations of this metal [14,16,17]. However, it is necessary to contemplate the environmental consequences of their medical usage and their further destination as a residue.

The physicochemical properties of AgNPs and the biological factors directly related to the in vitro, ex vivo, and in vivo models, considering the complexity of lung models, could provide a panoramic view of the potential environmental and human health risks. This review describes and integrates the different factors associated with pulmonary toxicity from exposure to AgNPs. Factors such as size, shape, coating, concentration, exposure time, cell type, and model-dependent cytotoxicity are highlighted to explain the cytotoxic and inhalation toxicity of AgNPs in several cellular and murine models. Finally, we expose the importance of the differential expression of genes as a tool to improve the prediction and interpretation of specific cytotoxic responses and the quantification of the damage itself. To the best of our knowledge, no other review has incorporated all these variables and parameters to study the results of the toxicity of silver nanoparticles in the different lung models.

## 2. Relation between Physicochemical Properties and Their Target Biological Models (PP × BM)

The relation between the PP of AgNPs and their target biological models (PP × BM) has been extensively studied. Within the most reported PP, we can find the size, surface functionalization, coating, redox potential, agglomeration, coating, and shape [2]. These properties exhibit different behaviors and degrees of biological interaction at the nanoscale than their bulk counterparts.

### 2.1. Size

AgNPs’ size directly impacts lung cellular cytotoxicity, inducing DNA damage, ROS production, mitochondrial dysfunction, lysosomal disruption, cell cycle arrest, apoptosis, and necrosis, among other effects on the cells in lung monocultures [11,15,16,17,18,19,20,21,22,23,24,25]. Particularly small-sized AgNPs (< 20 nm) are often related to a greater degree of toxicity [4,15,18,26]. Some lung cytotoxicity studies confirm the ability of small-sized AgNPs to breach across the intact cellular membrane. These dimensions allow the NP to pass across the cellular membrane by direct pore penetration or, by defect, improving the cellular uptake and translocation by endocytosis [17,18,19]. Even when AgNPs are not internalized, they can effectuate cytotoxicity by signaling pathway enrichment mediated by membrane protein receptors [20].

The transepithelial/transendothelial electrical resistance (TEER) assay is one of the most reliable techniques to identify cellular damage by monitoring tight junction dynamics in endothelial and epithelial models [21]. This model is an in vitro barrier model measurement technique based on ohmic and/or impedance evaluation in various frequencies [22]. This tool is helpful for the preliminary drug delivery and permeability studies of some lung barriers such as the blood–air, alveolar–epithelial, alveolar–macrophage barrier, and mucus barrier [23,24].

On the other hand, complex models such as co-cultures and 3D models provide access to apical and basolateral compartments, which can be seeded onto a permeable membrane [22,25]. Hence, the characteristics of physiology and functionality obtain relevance in vitro for further extrapolation to ADME in vivo and barrier integrity studies [22,23,27]. However, the results obtained should be considered carefully to avoid misleading interpretations. Zhang, et al. showed that lung co-cultures which could overcome 12 nm AgNPs’ cytotoxicity elicited better than monocultures. However, selective cytotoxicity to adenocarcinoma human alveolar basal epithelial cells (A549 cells) compared to their healthy counterpart BEAS-2B cells, is only obtained with the application of ultrasound [26]. Moreover, 10–20 nm polyvinylpyrrolidone-AgNPs (PVP-AgNPs) combined with hydra protein produced a similar result in 3D cultures of A549 exposed to them [27], in which viability was less affected than 2D monocultures of the same cellular line. In contrast, the ex vivo precision-cut lung slices (PCLS) model of C57/Bl6 female rats elucidates a more significant cytotoxic response than their in vitro human counterparts’ lung fibroblasts (HLF-1). Regardless of the size, 10 or 75 nm, AgNPs reduced the metabolic activity of PCLS and HFL-1 and modified the proteins in the extracellular matrix (ECM), promoting a pro-fibrotic response [28].

### 2.2. Surface Reactivity

The dimensions endow the nanomaterial with high biological reactivity due to the surface/volume ratio increase; this means more significant silver atoms and interaction with lung components on its surface [7]. Adding a capping agent significantly improved AgNPs’ surface reactivity and influenced surface charge, solubility, and hydrophobicity in a meaningful manner. Furthermore, surface properties directly affect the process of biocorona formation, pulmonary cell uptake, and lung biological barrier trespassing of AgNPs [29]. Hence, the importance of developing an extensive battery of bioassays to evaluate the degree of interaction exerted by the uptake of silver nanoparticles in living organisms has been potentiated and employed in recent years.

### 2.3. Agglomeration

Some studies report that the agglomeration status of AgNPs has a direct influence on the total cellular AgNPs. Ha, et al. reported that branched polyethyleneimine (bPEI)-AgNPs with smaller sizes increased their agglomeration, resulting in higher effective doses and cellular association than agglomerations bPBI-AgNPs with larger sizes [30,31]. Gliga, et al. demonstrated that five AgNP formulations (Citrate-AgNPs: 10, 40, and 75 nm; PVP-AgNPs: 10 nm, and uncoated 100 nm) exhibit some degree of agglomeration independently of the capping agent; showing higher levels of intracellularly AgNPs in healthy bronchial lung human cells BEAS-2B when the agglomeration rate increase [18]. However, several authors report the independency of intracellular uptake rate and cytotoxicity, elucidating a trojan horse mechanism for AgNPs within the lungs. It is imperative to mention that, in our opinion, the AgNP formulations’ response considered for the trojan horse mechanism are erroneously generalized, regardless of whether they have a coating agent or not, although the most are uncoated [32,33,34].

### 2.4. Ion Release

The cytotoxic effect of silver nanoparticles in mammalian pulmonary cells is often ligated to Ag^+^ ion release intracellularly so-called trojan horse mechanism. AgNPs in an aqueous solution interact with oxygen promoting the liberation of Ag^+^ ions, which, in turn, can interact with cell organelles and cellular membranes interfering with several biological functions [35]. It is also proved that inside the cells, AgNPs can suffer a series of biotransformations augmenting the bioavailability of Ag^+^, thus favoring the linking to other ions such as Cl^−^ and S^−2^, generating AgCl, AgS_2_, Ag_2_O, and Ag–Cysteine [36]. De Matteis, et al., demonstrated the intracellular release of Ag^+^ ions after AgNPs’ internalization, showing that the acidic lysosomal environment promotes particle degradation. However, the low citrate amount used to obtain the 20 nm AgNPs could explain the size increase observed in DMEM cell culture medium (~78 nm), probably for citrate substitution by proteins present in the culture medium [37].

### 2.5. Surface Functionalization

Surface functionalization is one of the essential features that can be manipulated to improve the biocompatibility and specificity of AgNPs. These provide AgNPs with inferred selectivity towards tumoral lung cells by interaction with specific tumoral receptors. This has been the case for many widely studied targets therapies in non-small lung cancer cells (e.g., EGFR and ROS1 mutations and ALK translocations) and several surface cell receptors in small cancer lung cells (e.g., PCRs, CXCR4, GLUT1, PETA/CD151, ALCAM/CD166, IGF1R and FGFRs, NCAM/CD56, RTK, ASCL1, SOX2, 4, and 11, OCT4, NANOG, PAX5, MYC) [38,39]. The surface functionalization of AgNPs is a critical factor that influences cellular uptake and their retention within the lungs [40]. Several agents can coat AgNPs to attack lung-specific tumoral receptors.

Based on the capping precedence, functionalized AgNPs can be subdivided into four vast families, namely: biogenic, phytogenic, polymeric, or in their absence, uncoated (Figure 1). The coating agents act as stabilizers preventing the agglomeration of AgNPs and could influence the uptake by healthy lung cells and, in turn, their cytotoxicity. In the recent investigation, it was clear that every coating agent produces a different level of cytotoxicity for lung cells, also modified by the variable model complexity (Table 1).

Surface functionalization also favors AgNPs’ interaction with the lung microbiome [61]. Depending on their coating, AgNPs could alter lung microbiome after instillation, producing a more pronounced the effect with citrate-AgNPs compared with PVP-AgNPs. The latter significantly reduce the inflammation produced by ovalbumin in BLAB/C mice and produce no adverse effect on non-sensitized mice. In these mice, the lung microbiome was altered by AgNPs increasing the abundance of Actinobacteria, Bacteroidetes, Firmicutes, and Proteobacteria [40]. On the other hand, AgNPs can interact with the lipopolysaccharides of the microbial wall present in Gram-negative bacteria through hydrogen bonds and electrostatic interactions. After exerting their antimicrobial power, AgNPs could generate a new coating with the lipopolysaccharides (AgNPs–LPS). The new coating favors an immune response through the interaction of LPS with the toll-like receptor 4 (TLR4) present to a greater extent in pulmonary macrophages [61]. This will generate its activation and therefore the production of cytotoxic inflammatory mediators such as cytokines, chemokines, and tumor necrosis factor-alpha (TNF α) which can exert lung adverse effects and even cancer in an indirect manner [62,63]. AgNPs–lung microbiome interaction is a very scarce topic on which further careful study is highly recommended. In this regard, the choice of AgNP coating has proven to be so important that if it is wrongly chosen it can lead to serious consequences in the lungs and the environment, even due to causes unrelated to the AgNPs–cell interaction.

#### 2.5.1. Uncoated AgNPs

Uncoated AgNPs are often considered the formulation with the highest toxicity. The aggregation tendency of uncoated AgNPs in solution and their relation to higher levels of cytotoxicity is well documented [18]. Some authors attribute these effects to promoting oxidative stress due to a reduced oxidant resistance caused by the lack of a capping agent (stable to oxidation). Moreover, uncoated AgNPs tend to dissolute and agglomerate faster in the biological medium [18]. The increased Ag release and agglomeration rates have been reported to cause severe cytotoxic effects in several lung cell models [43,64].

Ávalos, et al., assessed the toxicity of 4.7 and 42 nm uncoated AgNPs in primary cultures of human pulmonary fibroblasts (HPF), indicating an apparent size-dependent cytotoxic effect for the AgNPs at 4.7 nm [56]. Cellular damage exhibited as oxidative stress, gene upregulation, and G2/M phase cell cycle indicating mitochondrial disruption was found in organotypic-reconstituted 3D human primary small airway epithelial cell culture after inhalation of aerosolized 14 nm AgNPs [59]. Fizeşan, et al. [60] assessed the biological differences between 20 and 200 nm AgNPs in a complex 3D model representing the alveolar barrier. Regardless of the size, both AgNPs induced nuclear translocation of the transcription factor Nf-kB in endothelial cells at high doses, an essential marker in physiological respiratory diseases. The level of pro-inflammatory gene expressions such as IL-6 and IL-8 was also observed in a dose-dependent and time-dependent manner [60].

#### 2.5.2. Phytogenic AgNPs

Phytogenic AgNPs have gained much interest in the past few years by promising the reduction of the environmental footprint using natural bio-components in manufacturing [65]. Environmental sustainability is potentiated by employing nontoxic precursors and mild reaction conditions [66]. The most common precursors include polysaccharides, biodegradable gums, and phytoconstituents of some plants (biodegradable gum). Moreover, phyto-synthesized AgNPs (Ps-AgNPs) have shown essential effects on cancer cell inhibition without affecting healthy cells [67].

Ps-AgNPs independent of the original precursor yielded antitumor activity for every work registered in Table 1. Interestingly, these works are all related to the cytotoxicity evaluation of the A549 cell line. Several studies reported lung cell cytotoxicity in a concentration-dependent manner for small-sized [45,47,49,52] and big-sized [48] AgNPs. Some other authors disclose the time and size-dependent cytotoxicity for this particular cell line for medium-sized AgNPs [50,51]. However, only a few works compare healthy and tumor lung cells to describe a selective effect on malignant cells for medium- and small-sized AgNPs [28,56,60,61]. Hence, one thing is clear, Ps-AgNPs have an unselective impact on the cell cycle and cell death using DEGs dedicated to the apoptotic pathway such as p53, Bax, Bak, Casp3, and Cas9, among many others [53,59].

#### 2.5.3. Citrate-AgNPs

Citrate-AgNPs are stabilized by repulsion due to their negatively charged surface [68]. This negative surface can interact with metallic species in aquatic environments and modify its toxicity [69]. In biological media, citrate-AgNPs have also shown lower dispersion stability than PVP [70]. Citrate-AgNPs are the second most used and recommended coating agent due to their low cytotoxicity compared with extremely toxic uncoated AgNPs. However, as shown in Table 1, citrate-AgNP formulations lack cytotoxic selectivity. In fact, considering their effects on bronchial healthy lung cells, BEAS-2B was elucidated for small and big-sized AgNPs [14]. This goes along with the results found by Miyayama and coworkers [59], exhibiting toxicity for medium-sized AgNPs to small human airway epithelial cells. However, Schlinkert and colleagues describe the absence of cytotoxic damage for primary lung epithelial cells (NHBE) compared to BEAS-2B and A549, indicating an evident cell type-dependent toxicity [55]. This unselective cytotoxicity obtained with citrate-AgNPs can be attributed to their negative surface charge [71].

#### 2.5.4. Polyvinylpyrrolidone-AgNPs

Polyvinylpyrrolidone (PVP) medical usage was highly documented for various applications such as pharmaceutical, medicine, cosmetics, and food [72]. PVP can act as a stabilizer and reducing agent in the preparation of silver nanoparticles due to their amphiphilic nature and low toxicity [73]. Herein, AgNPs have been considered by different authors as the most stable formulation that can protect the nanoparticle from forming aggregates and reduce monodispersity [74].

A549 cells are the most reported cell line in the literature used to evaluate the cytotoxic effect of PVP-AgNPs. Briefly, the use of PVP-AgNPs of medium (40 nm) and large (80 nm) sizes in monocultures of A549 has shown size- and dose-dependent cytotoxic effects [75]. AgNPs were even visible at the structural level since cellular heterogeneity was observed in cells treated with AgNPs compared to the untreated population, attributing this effect to the Ag^+^ ion. Blanco and coworkers related the cytotoxic effects of 20 nm PVP-AgNPs to the genetic expression modification [41]. This formulation drastically decreased protein expression levels of p53, p21 MDM2, caspase 3, and MnSOD in a dose- and time-dependent manner. DEGs were an indicator of apoptosis, reduction of tumoral suppression, and mitochondrial ROS production, which was related mainly to the decrease in cell viability. Another example of small size AgNP toxicity reported by Rosário and colleagues shows a notable cell viability reduction produced by 10 and 20 nm PVP-AgNPs in a time-dependent manner. Besides, changes in the cell cycle indicate damage at the DNA level for the A549 cell line [42].

Comparative studies of the cytotoxicity of PVP-AgNPs in multiple lung cell lines have also been detailed. Comparing a four-cell line mosaic (A549, BEAS-2B, Calu-1, NCI-H358) to analyze the effects of a 23 ± 14 nm PVP-AgNPs formulation showed a decrease in cell viability in a dose and time-dependent manner. The results indicate a greater sensitivity for Calu-1, followed by BEAS-2B, A549, and NCI- H358, which proved to be the most resistant [43]. The effects at the cell cycle level indicated arrest in G2-Phase for Calu-1 and A549 and arrest in the S-Phase for BEAS-2B, while for the NCI-H358 line, there were no significant changes, again indicating that it was the most resistant. ROS production was observed in all lines except for NCI-H358; meanwhile, BEAS-2B and Calu-1 showed the highest increase in protein oxidation. Finally, the levels of ATP production exhibited a dose-dependent decrease to a greater extent for the Calu-1 cell line, indicating that the Calu-1 line and the NCI-H358 line are at opposite extremes of sensitivity for the different studies, making explicit a cell type-dependent cytotoxic effect [76].

Similarly, PVP-AgNPs have been shown to have a selective antitumor effect in 3D models. Haase, et al. performed an in vitro cytotoxicity assay using an EpiAirway^™^ 3D human bronchial model and three variants of AgNPs (Citrate-AgNPs 50 nm, PVP-AgNPs 50 nm, and PVP-AgNPs 200 nm). PVP-AgNPs 50 nm and 200 nm showed a non-cytotoxic nor genotoxic effect for this specific model, while citrate-AgNPs demonstrated the opposite results [77] (Figure 2). Lee, et al., identified reduced cytotoxicity of PVP-AgNPs 10–20 nm in combination with a hydra protein in a 2D and 3D A549 model. Cytotoxicity in the form of apoptotic and necrotic cells was observed in the 2D cell culture in minor concentration than in the 3D model; this exhibits that the 3D model is an accurate model to simulate the in vivo conditions since more factors are involved [27].

## 3. Biological Factors Involved in AgNPs Cytotoxicity

### 3.1. Cell Type

This section will focus on how AgNPs interact with the distinct lung cell lines by meaning cell-specific toxicity, the nature of the biocorona formation, and their implications for the respiratory system.

### 3.2. Pulmonary Cell Type

#### 3.2.1. Bronchial Epithelial Cells (BECs)

Bronchial epithelial cells can be subdivided into three principal categories based on their biochemical and structural characteristics: basal, ciliated, and mucous cells [78]. Basal cells can be found all along the large and small airways, diminishing in total quantity with the periodical decrease of the airway size. They are only expressed on hemidesmosomes [78]. On the other hand, ciliated cells compose the vast majority of bronchial epithelial cells (around 50%), which are in charge of the mucociliary escalator right function. Finally, the mucous or goblet cells act as mucin glycoprotein producers to trap foreign agents in the lung lumen. The airway epithelium lines, the trachea, and alveoli by pseudostratified tissue contain a single layer of cells with different forms and a column of distinct size columnar cells forming a tight junction around the lateral apices of columnar epithelia [79].

Normal human bronchial epithelial cells (NHBE), either primary or immortalized, are the conventionally most used model for AgNPs bronchial damage evaluation in vitro, as shown in Table 1. BEAS-2B (normal human bronchial epithelial cells) and primary NHBE are the primary targets for cytotoxicity in bronchial cells. Interestingly, despite the differences in both epithelial cell types, these models have proven the absence of significant cytotoxicity in several studies [9,66,80]. However, primary NHBE was more sensitive to many functionalized AgNPs than immortalized cell line BEAS-2B. The observed response is attributable to the capacity of BEAS-2B cells to form tight junctions under submerged conditions. On the other hand, primary NHBE cells need more adaptation time [14]. 

In contrast with NHBE, very little has been reported about the antiproliferative effect of AgNPs on bronchial tumoral cells (BTC). As the only example in the literature, Holmila and collaborators [43] found that Bronchioalveolar Carcinoma epithelial human cells (NCI-H358) were exposed to 1–10 μg/mL of commercially available PVP-AgNPs for 24, 48, and 72h showing no cytotoxic effects. NCI-H358 exhibited AgNP-resistance compared with tumor and non-tumor epithelial cells A-549, Calu-1, and BEAS-2B. The AgNP-resistance of NCI-H358 is associated with the lowest level of mitochondrial ROS content compared to their counterparts. Redox proteomics analysis confirms the differential toxicity and the mitochondria as the main target of AgNPs, confirming the importance of quick cell metabolism regulation capacity of cells as a response to environmental stressors [76]. These results open a hot spot by validating the antiproliferative activity of AgNPs in vitro on bronchial tumor cells to determine the cytotoxic selectivity of AgNPs.

#### 3.2.2. Alveolar Epithelial Cells

One of the key and most investigated models in lung toxicology is the alveolar epithelium, for a number of reasons. The alveolar epithelium comprises two types of cells: alveolar type I (AT1) and alveolar type II cells (AT2). AT1 covers more than 95% of the alveolar surface, and they are essential for the air–blood barrier and gas exchange process of the lung [16]. AT2 cells act as progenitor cells for types 1 and 2, mediating damage produced in AT1 cells and protecting the alveoli surface by generating pulmonary surfactant (PS) [81]. In addition, AT2 has glutathione to provide the antioxidant defense against ROS and provide immune defense through the presence of alveolar macrophages within the AT2 epithelia [82]. These features make alveolar cell models a primary screening model in lung toxicology testing.

Alveolar cells are the most reported cell lines in lung toxicity measurements, representing almost 85% of the reports from the past five years compiled in Table 1. This fact is why the human alveolar adenocarcinoma cell line (A549) has become the gold standard model for lung toxicology. The A549 cell line is derived from AT2 but maintains some unique features of AT1, such as the presence of a caveolin-1 specific biomarker, being one of the only commercially available cell lines of this type [83]. A549 cells generate confluent monolayers with AT2 functions, making them suitable as a model for the metabolism and liberation of drugs in the alveolar epithelium.

A549 cells have emerged as a well-established drug screening model for a long time. However, in recent years, this particular model has been employed as a target for several AgNPs with different PP, resulting in a decrease in tumor cell viability [53,54,61,72]. Nevertheless, some authors manifest that AgNPs did not significantly decrease A549 cell viability, even with AgNPs with similar properties [55]. These findings have raised the complex models needed to better understand lung cytotoxicity in pulmonary models and obtain physiologically relevant results as in vivo. Herein the biggest challenge is incorporating the different components of the lungs into the same model (e.g., pulmonary surfactant and immune cells).

#### 3.2.3. Macrophages

Alveolar macrophages (AM) and interstitial macrophages (IM) make up the two types of existent macrophages in the lungs. Alveolar macrophages are the first line of defense against airborne NPs, pollutants, and pathogenic agents. They are responsible for the clearance of debris and the recycling of PS. On the other hand, interstitial macrophages are considered the second line of defense against dangerous agents in contact with the lungs. IM has vital immune roles, is responsible for maintaining lung homeostasis, and mediates airways’ allergic reactions [80]. Regardless of the macrophage type present in the lungs, they exert phagocytic activity under physiological conditions, play a significant role in adaptive immunity, and control inflammation by producing cytokines [84,85]. Therefore, alveolar macrophages have been incorporated into diverse lung cell cultures, providing new insight into drug mechanisms.

The air–blood barrier is mainly conformed by alveolar macrophages and alveolar epithelial cells in vivo. The incorporation of macrophages has improved the physiological relevance of lung cultures in vitro. A diffusional barrier interpretation by the co-culture of human alveolar epithelial cells (hAELVi) and macrophage (THP-1) exhibit resilience to AgNPs’ cytotoxicity than cell lines evaluated by their own with a size-dependent behavior regardless of the coating agent, PVP, citrate, or uncoated. Interestingly, no cytotoxic difference was found in PVP- and citrate-AgNPs of 10 nm. However, it is important to note that the coating agent percentage on assessed AgNP formulations is lower than 5% in all cases, which could help to explain the “Trojan horse” mechanism proposed [18]. The same conclusions were obtained from a 3D model representing the alveolar barrier, showing a decrease in cell viability after 6 h of exposure to 20 and 200 nm uncoated AgNPs with a system recovery of 24 h [60]. Compared to monocultures, increased resilience found in complex lung models could be attributed to the bidirectional communication between lung alveolar/bronchial cell lines and immune cells. The response can be either by direct contact or by the interference of soluble mediators. Despite the rise in the resistance to acute cytotoxicity of the system, an increase in the total cytokine production indicates immune or inflammatory responses, also related to common factors [18]. Even considering in vitro the relevant biological barriers mentioned above, it is necessary to consider the biotransformation of AgNPs mediated by the adhesion to the surface of NPs of excreted products by the different cell types, a process known as a biocorona formation.

### 3.3. Biocorona Formation

NPs can interact with their surroundings due to their high surface reactivity, producing the adhesion through intermolecular forces, electrostatic interactions, or covalent bonds of protein, metabolites, lipids, and nucleic acids [2]. The process, known as biocorona formation, is influenced by the physicochemical properties of NPs and the biological environment [86], water repulsion forces, and favorable entropy variation [87]. Biocorona structure could be divided into the hard and soft corona. The former refers to the strongly bonded layer of molecules by proximity to the AgNPs surface. The latter is related to the exterior weakly bonded molecules constantly renovated while traveling into a changing biological media.

The AgNPs can link to pulmonary surfactant and mucus in the deep lungs in the respiratory system. Both PS and mucus conformed by protein and lipids are responsible for clearing and selective internalization of foreign agents. While mucus act as a protective epithelial layer, PS diminishes alveolar surface tension of the epithelial air interphase. Phospholipid dipalmitoyl-phosphatidylcholine, and the surfactant proteins A, B, C, and D are the most prominent agents present in PS. Protein A is the most abundant and responsible for surfactant homeostasis and immune response regulation; surfactants B and C are in charge of the laminar transformation process and their dissemination over the alveolar epithelium, and finally, protein D binds to several microorganisms and lymphocytes [88]. The composition of AgNPs-corona highly depends on the surface charge, hydrophobicity, and coating-molecule affinity [84]. Surface modification of AgNP will affect their overall behavior, e.g., colloidal destabilization, opsonization, aggregation, increase or decrease in circulation times, phagocytosis rate raised by macrophages, bioaccumulation, and cytotoxicity [89].

Moreover, it has been proved that AgNPs-PPs regulate the translocation across the PS monolayer and the formation of lipoprotein corona [90]. AgNPs can interact with sulfur and selenium tissue forming particles within the lungs of animals and humans [91]. Davidson, et al. explain the process of AgNPs biotransformation on rats in vivo by inhaling AgNPs 20 and 110 nm after seven days of exposure. The authors found that AgNPs turned into smaller or, in their defect, zeolite-like nanomaterials by dissolution confirmed by x-ray absorption methods [24].

It has long been stated that the nature of biocorona highly determines the level of uptake in lung cells and, therefore, directly affects the observed toxicity [92]. Barbir and co-workers working with albumin-, metallothionein- and PVP-coated AgNPs reported that protein corona affects biodistribution with a gender dependence and modulates the redox response and genotoxicity observed in the exposed tissue, as discussed in the next section [92].

### 3.4. Gender

Although very few reports exist, in vivo studies have shown the gender-dependent toxicity of AgNPs in the lungs could be related to hormone signaling, lung physiology, and respiratory immune function [93]. Ovarian hormones are usually associated with a pro-inflammatory response in the lungs [94], such as airway inflammation in asthma [95]. PVP-AgNPs and TRF-AgNPs (TRF = transferrin) administered to intact and gonadectomized B57Bl/6 adult mice produce oxidative damage in the lungs through ROS overproduction and GSH depletion on intact female and gonadectomized males compared to intact males and gonadectomized females. These results suggest a strong influence of serum progesterone levels on oxidative lung damage [39]. Gender-related differences in the biokinetic profile in blood and lung distribution have been found after intravenous administration of 15 nm PVP-AgNPs at concentrations of 7.5 to 120 mg/Kg of body weight in ICR mice. The half-lives of elimination show higher retention of AgNPs in females compared with male mice, with 29.9 and 15.6h, respectively [96]. Furthermore, 10–15 nm MT-AgNPs (MT = metallothionein) also produces a higher accumulation of silver in the blood of females compared with male Wistar rats after intravenous administration of 1mg Ag per Kg bodyweight. MT-AgNPs produces DNA oxidative damage after 1h of i.v. administration in blood, liver, and kidney female cells. PVP-AgNPs and albumin-AgNPs of the same size produce a similar response but lower effect [97]. Lung oxidative damage was also observed on ICR mice administered with 20 nm uncoated AgNPs and PVP-AgNPs by gavage at a dose of 10–250 mg/kg body weight per day for 28 days. No sex-differentiated responses could be observed as both are included in the same experimental group. Still, interstitial inflammation, bronchial tissue necrosis, and foam cells appeared in the alveoli [98]. On the other hand, Sprague Dawley rats exposed to sub-chronic doses of 18–19 nm AgNPs for 6 h/day, 5 days/week, for 13 weeks in a whole-body inhalation chamber show mixed inflammatory cell infiltrate, chronic alveolar inflammation, and small granulomatous lesions. Despite no gender-related differences found in silver accumulation in lungs, female kidneys contain two to three times more silver than male kidneys [99]. Sprague Dawley females rats exposed to 18 nm AgNPs at concentrations of 0.7–2.9 × 10^6^ particles/cm^3^ for 6 h/day in an inhalation chamber for 90 days exhibit dose-dependent lung inflammation [100]. However, increasing time and concentration exposure shows an exposure-related lung function decrease in males compared with female rats exposed to AgNP even after 12 weeks of recovery [101]. Although the exact mechanism of gender-dependent toxicity is still unclear, some effects can be attributed to DEGs between genders and rat strains generating new measurable parameters while evaluating AgNPs cytotoxicity.

### 3.5. Cytotoxic Response-Dependent on Extrinsic Factors

#### 3.5.1. Time of Exposure

Time-dependent cytotoxicity is one of the most evaluated factors in AgNPs biological studies. The different exposure time to AgNPs directly impacts several parameters such as pH, internalization, and dissolution of NPs into Ag^+^ [102]. The release of silver ions has been reported as the primary AgNP toxicity mechanism due to their capacity to form coordination complexes with thiolate groups and produce inflammation, dysfunction of several organelles, metabolic alteration, and DNA damage, among many others [103]. However, pulmonary cytotoxicity of AgNPs is not only related to ions. As stated before, NPs-biocorona stability with lung components is time-dependent. The protein corona provides the necessary tools for AgNPs to trespass the alveolar–capillary barrier and distribute to remote organs generating chemotaxis and other pathophysiological effects on the lung and cardiovascular system [104]. Citrate-AgNPs increased cyto- and genotoxicity in a time-dependent manner through p53 up-regulation, leading to apoptosis on A549 cells. Modification of citrate-AgNPs with lactate or a 12-base oligonucleotide trigger high DNA damage [105]. Additionally, 24 nm PVP-AgNPs induced in vitro lysosomal injury, mitochondrial membrane potential decrease, and oxidative damage leading to A-549 exposed cells autophagy and mitophagy in a time-dependent manner [106]. Similarly, 40–90 nm AgNPs reduce cell viability and mitochondrial membrane potential leading to A-549 cell death through ROS-dependent and ROS-independent pathways [107]. In vivo experiments show that inhaled 13–16 nm AgNPs increase silver levels on Brown–Norway and Sprague Dawley rats’ lung macrophages, indicating lung toxicity persistence even after 7 days of exposure and the absence of airway luminal inflammation [108].

#### 3.5.2. Concentration/Dose

Concentration is another highly reported factor influencing overall AgNPs cytotoxicity is concentration. Cell viability is often reported to decrease linearly to the increase of AgNP concentration [18,40,41,83,104]. The cell viability decrease can be associated with ROS overproduction, LDH membrane leakage, and mitochondrial transmembrane potential disruption for several in vitro and in vivo models [40,47,55,60,74,109,110,111]. Different concentrations are reported to exert double-strand DNA break, inflammation, and lung epithelium damage and impact cell viability, immune response, and bioaccumulation [14,18,28,41,42,59,83,103,105,112,113]. The aforementioned cytotoxic parameters regulate numerous steps in the intracellularly signaling cascade by specific transcription factor overexpressing and increasing bioactive molecules such as cytokines in a dose-dependent manner [40].

#### 3.5.3. Model

Model-dependent cytotoxicity is a scarce topic in nanotoxicology. Despite being the most relevant factor in determining the cytotoxicity of AgNPs, minimal investigations were carried out to elucidate these effects. To the best of our knowledge, no one has provided a focal review of this parameter. It is clear to us that the complexity of divergent in vitro models such as monocultures, co-cultures, and 3D cultures will considerably affect the cytotoxic response to AgNPs, besides the number of variables that can be measured for every particular model. The increasing complexity of the model offers a robust and more accurate representation capability of the in vivo microenvironment. In this work, we establish the basis of the lung cytotoxicity model dependence while remarking on the importance of its study.

## 4. Nanotoxicity Models to Evaluate Lung Cytotoxicity

Lungs are the first organ to face constant exposure to nanoparticulate matter; therefore, different biological models have been used to evaluate their cytotoxicity over the past ten years. Among the most studied, we have in vitro models, which represent an economic, easier-to-replicate, and ethically kinder alternative to in vivo models [114]. These particular models exploit cell culture systems to carry out a wide array of experiments with specific endpoints to predict possible hazard effects of the AgNPs’cell interactions [29]. In vitro cell cultures can be subdivided into three principal branches based on their complexity level: monocultures (one type of cell), co-cultures (two or more types of cells cultivated on the same media), and 3D cultures (two or more types of cells grown on a matrix) [115]. The added variables could better simulate the in vivo microenvironment, increasing cell–cell interactions and providing access to nutrients, oxygen, and metabolites [109]. The insight provided from in vitro models can be further translated into in vivo situations after incorporating different cell types and structures, highlighting the need to continue using both models to predict the cytotoxicity of new therapeutic agents or the toxicological effects of nanoparticulated matter.

Different variables can affect the complexity of in vitro experimental design building and its possible effectiveness as a predictive lung toxicity model. The first factor to consider is the cell lineage selection: primary, pluripotent, or transformed. All three have qualities and limitations that can be exploited to obtain reliable models. Primary cell lines are isolated directly from animal tissue or tissue from human donors, conserving their natural morphology and characteristics, thus making in vivo mimicking more suitable [116]. However, isolation techniques require some regulation, are relatively expensive, and must be carried out under axenic conditions. Besides, the primary cell life span on culture is limited, so their self-renewal potential is reduced.

On the other hand, working with cell lines is more accessible, cheaper, and the cell culture is maintained for longer. Still, they present some problems such as genotype and phenotype variations, lack of relevant biomarkers, and a lower physiological relevance and sensitivity to drugs than primary cultures [110]. Nevertheless, these cell types can be used to prepare co-cultures that resemble more complex culture systems that better represent the biological barriers within the lungs.

As we become more and more exposed to hazardous nanoparticulated matter in the air, our lungs evolve and develop a series of complex barriers to protect us. These barriers are distributed all along the respiratory tract. Upon inhalation, until the gas exchange surface on the alveoli, external agents must overcome three principal obstacles: physical, biochemical, and immunological barriers. Physical barriers are represented by 23 tubular bifurcations gradually reducing their ratios and generating complex branched structures governed by forces of diffusion and deposition (e.g., inertial impaction, sedimentation, Brownian diffusion, and electrostatic deposition) [111].

In the case that AgNPs are deposited into the respiratory surface, they encounter larger molecular weight glycoproteins ranging from 200,000 to 3 million Daltons, better known as mucus on the upper and conductive airways, which conforms to a bilayer barrier surrounding ciliary cells [117,118]. In contrast, if AgNPs avoid deposition by the conciliary escalator, they will reach the alveoli, where a protective 0.2–5 μM layer of pulmonary surfactant can bind to the AgNPs’ surface by corona formation processes leading to excretion, ingestion, or translocation. The fluid secreted by type II cells is composed of lipids and hydrophobic surfactant proteins [119]. The last line of defense contains type I and II pneumocytes monolayer in the epithelial tissue of the alveoli and a group of heterogeneous macrophages distributed on the respiratory surface. These cells are responsible for optimizing gas exchange with the atmosphere, producing lung surfactant, and performing homeostatic functions for correct tissue functioning [120]. It is expected that the complex barriers and systems of the lungs are difficult to emulate in vitro. However, significant advances have been reported in recent years to bridge these existing gaps with their in vivo counterparts, providing new insights for the future of in vitro lung nanotoxicity research.

### 4.1. Lung Mono-Cultures

The easiest and cheapest way to study lung toxicity of nanomaterials relies on cell monocultures. Monoculture techniques consist of a single cell-type thin monolayer formed over a plastic plate under axenic conditions enriched with media and essential nutrients that allow its proper growth. Cells can be isolated directly from a human or animal donor or by employing immortalized cell lines. Mono-culture cytotoxicity assays help predict single-cell responses to stimuli, including cell death pathway, inflammatory response, epigenetic and phenotypic changes, membrane integrity, mitochondrial protein oxidation, dysregulation of genes, and oxidative stress. However, it is essential to note that these simple models fail when trying to emulate more complex interactions such as cell–cell and cell–matrix signaling. Therefore, an appropriate experimental design must contemplate the cell line selection based on the biological endpoint, such as cytotoxicity, pro-inflammatory response, oxidative stress, and/or genotoxicity.

### 4.2. Lung Co-Cultures

The co-culture technique can be seen as the generation of cell cultures of two or more types of cells in a plastic dish supplemented with the essential nutrients for life. Cell co-cultures can mimic in vivo signaling responses more than mono-cultures. The possibility of having two or more cell types coexisting in the same media provides new measurable parameters based on cell–cell interactions and paracrine signaling by dissolution factors. Since pulmonary drug administration studies are usually intended to attack biological barriers, the co-culture complexity level will allow the air–blood barrier representation to study the effects of the nanoparticle when crossing or trespassing in the lung [121]. However, these added variables often represent low reproducibility rates and increase the time and difficulty of the experimental procedure. Another limiting factor to the physiological relevancy of this model is the lack of an extracellular matrix that provides a continuous series of nutrients and a greater degree of intercommunication, which supports the 3D model’s development interest. 

### 4.3. Lung 3D Cultures

3D cellular architecture is characterized by the simulation of in vivo microenvironment in vitro. The most relevant factor to consider in this model is the extracellular matrix addition (e.g., Matrigel, collagen, scaffold) that provides a three-dimensional structure and facilitates cell growth and adhesion [110]. The new cell–matrix interactions can be implemented to develop physiologically relevant in vitro tumor models for anticancer drug development [13]. Extracellular matrix (ECM) supplies oxygen and metabolites variable access; hence an increment in overall resilience to AgNPs’ cytotoxicity is observed on complex models. One good example of this is the representative models of lung tissue using primary human small airway epithelial cells (HSAEpCs) grown in fibronectin and collagen-coated chitosan scaffolds to study various respiratory diseases such as influenza virus [122]. Besides all the advantages of 3D models, it must consider that these are built based on cell lines avoiding primary cultures due to the uneasy and expensive design of the culture. This breach must be further studied to establish a true representative in vitro model of the lung conditions in vivo.

### 4.4. Ex Vivo Lung Models

The inability to fully replicate the human parenchyma makes the ex vivo technique, including precision-cut slices (PCS), suitable for studying respiratory responses while testing new drugs and compounds. Sauer and co-workers report the use of rat precision-cut lung slices (PCLS) to evaluate cytotoxicity, apoptosis, oxidative stress, and inflammatory response of sixteen OECD reference NMs. Among them, NM-300K, a <20 nm AgNPs dispersed in an aqueous solution containing polyoxyethylene glycerol triolate and polyoxo ethylene (20) sorbitan mono-laurate (Tween 20) as capping agents. Rat PCLS exposed to 12 μg of NM-300 K/cm^2^ of tissue for 24 h exhibit tissue destruction, severe loss of protein content in the BCA assay, condensed nuclei, vacuolated cytoplasm, and particles in macrophages and free in the alveolar lumen. Observed damage was associated with the silver ion shedding, probably by the low efficacy of the coating agents to stabilize the AgNPs formulation [123].

On the other hand, the exposure of rat PCLS to 70 nm PVP-AgNPs for 4 and 24 h shows only a slight cytotoxic and no pro-inflammatory response compared with uncoated ZnO-NPs, which showed a strong cytotoxic response associated with Zn^+2^ ion release. PVP-AgNPs evaluated in this work were found mainly in the cut surface without a significant amount within the tissue slide [124]. The effect of the coating agent and exposure time is evident by analyzing the results of murine PCLS exposed to 10 and 75 nm citrate-AgNPs at concentrations of 2 and 10 μg/mL for 24, 48, and 72 h. The exposed tissue significantly reduces metabolic activity, consistent with in vitro results obtained with HFL-1 cells. The immunomodulatory response shows that both citrate-AgNPs, 10 and 75 nm, may induce a concentration- and time-dependent pro-fibrotic response in human lung fibroblast but not size-dependent behavior [28]. Physicochemical parameters and exposure time must be considered in the experimental design for cytotoxic and immunomodulatory response evaluation on this model, showing the potentiality and limitations of the ex vivo assays.

### 4.5. In Vivo Lung Models

In vivo toxicological studies are the previous step before clinical trials. The most common experimental models are Balb/C and C57BL/6 mice and Sprague Dawley and Wistar rats (Table 2). The assays performed with the animals involve toxicity evaluation directly on the whole organism besides analyzing data from isolated tissues, organs, or cells. In vivo approaches have advantages and disadvantages; however, the combination of in vitro and in vivo approaches provides a better understanding of the toxicological profile of the evaluated substance.

In vivo assays provide meaningful information that cannot be obtained from in vitro cytotoxicity experiments. An example could be pathological changes such as thickening of interstitial tissues, focal interstitial pneumonia, lung inflammation, and lung fibrosis found after prolonged exposure to AgNPs [20,125,126]. However, cytotoxicity parameters obtained from in vitro assays, discussed in the above sections, i.e., oxidative stress, inflammation, mitochondrial fission, ion flux dysfunction, cytoskeleton damage, disruption of protein expression profile, and DNA double-strand breaks, lead to an outstanding in vivo experimental design.

The relevance of dose-dependent response becomes evident working with the whole organism. While some concentrations may induce low to moderate toxic effects, exposure to higher doses can result in pathological changes at the lung tissue level [127]. An interesting example of in vitro results to illustrate the interactions between AgNPs and alveolar macrophages upon in vivo exposure was provided by Liu and co-workers. In this work, intratracheal instillation of 20 and 110 nm AgNPs at a concentration of 0.5 mg/Kg body weight produces different scenarios according to the uptake amount of the alveolar macrophages functionality isolated from broncho-alveolar lavage fluid (BALF). The minimal intake shows no changes compared with the control, low to medium uptake affecting the cytoskeleton structure and is responsible for cellular stiffening, and finally, the uptake of higher amounts of AgNPs eliciting ROS overproduction and alveolar macrophages action leading to disintegration of actin network and softening of the cellular mechanics [128].

Time-dependent cytotoxicity of AgNP exposure, independently of the administration route, can be measured by assessing BALF at different periods. Long-term effects produced at the DNA level, reversibility of the inflammatory effect, and the Ag accumulation and biodistribution can be evaluated [112]. Dziendzikowska, et al. [129] described a size- and time-dependent behavior for Ag accumulation of the lungs after intravenous (IV) administration of 20 and 200 nm AgNPs to Wistar rats. 

**Table 2 nanomaterials-12-02316-t002:** Observed outcomes after in vivo administration of several AgNP formulations by different administration routes.

AgNPs Coating	Size (nm)	Dose (mg/kg BW)	Time of Exposure	Model	Observed Outcome	Ref
PVP	10–30	Daily i.p. of 0.25, 0.5, 1	9 d	Male Balb/C mice	Toxic damage in major organs at all doses (lung, liver, spleen, kidney, heart, brain, and testicles) Dose-dependent toxicity on the lung. Thickening of interstitial tissues and focal interstitial pneumonia (0.5 mg/kg BW). Significant interstitial pneumonia with massive cell infiltration and interstitial hemorrhage (1 mg/kg BW).	[127]
PVP	25	Final dose 0.02 using inhalation chamber	Exposure to 0.7 mg/m^3^ AgNPs for a half-hour every day until 45 days	Male C57BL/6 mice	Cell cycle arrest in the G2/M phase Upregulation of COX2/PGE2 intracrine pathway Accelerate lung cellular senescence Cause mild fibrosis.	[130]
PVP	50 and 200	3.75, 75, 150, 300 μg	3 and 21 d	Female Wistar rat	Dose-dependent toxicity. DNA double-strand breaks. Damage to alveolar macrophages and endothelial cell destruction Lung inflammation.	[40]
PVP and citrate	20, 60 and 100 nm	10 μg Ag/mouse	4 and 24 h	Male ICR mouse	IL-1β and neutrophils in BALF, lung inflammation but do not indicate if PVP- or citrate-AgNps produce it. Size-dependent toxicity for citrate-AgNPs.	[131]
Citrate	20 and 110 nm	0.5 mg AgNPs/kg BW	24 h	Sprague Dawley Rats	Size-dependent uptake and toxicity. Ion flux dysfunction, ROS production. Uptake-dependence produces cytoskeleton rearrangement, stiffening of mechanics, and cytoskeleton damage that softens the mechanical profile.	[44]
Citrate	20 and 110	Single n.a. 7.2 and 5.4 mg/m^3^	6 h	Male Sprague Dawley rats	Presence of silver in tissue macrophages obtained from BALF, 56 days post-exposure. AgNPs are predominantly localized within the lung’s terminal bronchial/alveolar duct junction region associated with extracellular matrix and within epithelial cells.	[128]
Citrate	20	o.a. 0.25	24 h	Male mice CBA/J, C57L/J, MRL/MpJ, NOD/ShiLtJ, NZB/BlNJ, NZO/HlLtJ, NZW/LacJ, PL/J, PWD/PhJ, PWK/PhJ, TALLYHO/JngJ, WSB/EiJ, BALB/cJ, BTBRT + tf/J, C3H/HeJ, C57BL/10J, DBA/2J, FVB/NJ, SJL/J, SM/J, SWR/J, 129S1/SvImJ, A/J, AKR/J, and C57BL/6J.	Strain and treatment-dependent in neutrophils in BALF with the exception of SWR/J, DBA/2J, and SM/J. Lung inflammation	[132]
Citrate	20 nm	Single IV 5	1, 3, and 5 d	Male Sprague Dawley rats	Time-dependent Ag accumulation in the lung. Ag^+^ accelerates the dissolution of citrate-AgNPs by MT overexpression.	[133]
Citrate, octreotide (OCT), and Citrate/ OCT/ alginate (ALG)	22.77 ± 1.1, 78.77 ± 2.3, and 155.99 ± 5.2 nm	Nebulization of 1.27 at a rate of 5 mL/h for 3d (10h/d)	3 d	Male and female Sprague Dawley rats	AgNPs surface modification with OCT and ALG favors AgNPs accumulation in the lung and enhances interaction with somatostatin receptors (SSRT)in tumor cell lines.	[134]
ND	14–15 nm	0.05, 0.12, and 0.38 mg/m^3^ (Low, medium, and high dose respectively) in an inhalation chamber	6 h/day, 5 days/week for 12 weeks	Male and female Sprague Dawley rats	Accumulation in greater quantity in the lung, and in a dose-dependent manner in the liver, kidney, blood, vessel, eye, and testicle. No effect on the brain High amounts of silver were maintained after 12 weeks in liver, vessel, and eyes	[101]
ND	18–19 nm	0.049, 0.133 o 0.515 mg/m^3^ (Low, medium, and high dose respectively) in a whole-body inhalation chamber	6 hours/day, 5 days/week for 13 weeks	Male and female Sprague Dawley rats	Accumulation in greater quantity in the lung, liver, vessel, kidney, brain, and olfactory bulb Accumulation in a dose-dependent manner 0.7, 1.8, and 4.3 μg silver/kg dry weight of tissue in blood	[99]
ND	15 nm	0.133 mg/m^3^	Laminar horizontal flow, ventilation exchange rate of 20 times/hr) for 6 hr	Female Fischer 344 rats	Accumulation in greater quantity in the lung, nasal cavities, lymph nodes associated with the lungs, and blood. Low in heart, liver, blood vessel, kidney, and brain. Recovery 7 days after exposure.	[125]
ND	18.1–19.6 nm	0.031, 0.082, 0.116 g/m^3^	6 h/day, 5 days/week for 4 weeks in a nose-only inhalation chamber	Male Sprague Dawley rats	Accumulation in lung with a recovery of half the day after 14.7, 6.4 and 1.6 μg silver/kg dry weight of tissue in blood, followed by a low elimination phase of 60 to 100 days	[126]
ND	20 and 200	IV single dose to the tail vein of 5	24 h, 7 and 28 d	Male Wistar rats	Time-dependent change in concentration of silver in the liver, spleen, kidneys, lungs, and the brain. The highest amount of silver in the lungs was observed after 7 days. Individual AgNPs and AgNPs cluster were found within lung macrophages attached to the alveolar wall and inside the interstitium. AgNPs accumulated in the cytoplasm, mitochondria, and nucleus.	[129]
ND	8–22	Daily i.p. of 0.01	36 d	female severe combined immunodeficient (SCID) mice	Apoptosis. Significant tumor growth decrease after 36 days of treatment. No toxicological effects studied.	[135]
ND	27.9–33.4 and 57.3–33.4	* More information in the paper	40 min	Rat (no defined strain or sex)	Neutrophil increase in BALF with a size and dose-dependent response. Lung inflammation.	[136]
ND	20 nm	i.i. 50 μg AgNPs/rat	7 and 28 d	Male Sprague Dawley rats	Lung parenchyma injury, alveolar collapse, parenchymal fibrosis. Partial recovery after 28d but persistence of inflammatory/fibrosis response.	[137]
ND	10–20	i.i. 200 μg per rat	(1) Once a day for 7 days (2) Single intratracheal instillation	Male Sprague Dawley rats	Enhancement of oxidative stress, mitochondrial dynamic imbalance. Thickening of the alveolar septa, accumulation of macrophages in the alveoli, formation of pulmonary bullae and pulmonary consolidation, the disintegration of the mitochondrial cristae, and swelling of the mitochondria.	[138]

BW = bodyweight; i.i. = intratracheal installation; i.p. = intraperitoneal injection; IV = intravenous injection; n.a. = nose aerosol; o.a. = oropharyngeal aspiration; * details of complete admininistration scheme could be consulted in reference [136].

The highest concentration of silver in the lungs was observed after seven days of administration and decreased over time; however, no coating agent was reported. On the other hand, a single nose aerosol administration of 20 and 110 nm citrate-AgNPs at 5.4 and 7.2 mg/m^3^ shows silver accumulation on BALF macrophages, particularly with the smallest nanoparticle, even 56 days post-exposure [139].

Scoville and collaborators developed a highly illustrative work determining the lung inflammation and toxicity caused by citrate-AgNPs of 20 nm at a dose of 0.25 mg/Kg of bodyweight by oropharyngeal aspiration, producing lung inflammation identified by neutrophils in BALF. The damage is identified in 22 of the 25 mouse strains used in that document with a strain-dependent behavior; besides, three promising candidate genes were identified as lung inflammation biomarkers. Nedd4l (neural precursor cell expressed developmentally downregulated gene 4-like; chromosome18), Rnf220 (Ring finger protein 220; chromosome 4), and Ano6 (anocatmin 6; chromosome15), for which mRNA levels were inversely correlated with AgNP-induced lung inflammation [132].

The stability provided by the capping agent to AgNP formulations directly influences the produced toxicity. The biodistribution and speciation study of 20 nm citrate-AgNPs after single intravenous administration of 5 mg/Kg body weight to male Sprague Dawley rats shows that most silver appears as Ag(I). Interestingly, the same group reported that AgNPs found in the lung and liver maintained their original size without evidence of smaller particles, suggesting the unlike reduction of Ag^+^ to AgNPs within the cellular environment to produce new AgNPs of different sizes. Also, Ag^+^ contributes to the dissolution of AgNPs by inducing metallothionein (MT) overexpression [133]. The surface functionalization of AgNPs with specific somatostatin receptor (SSTRs) binding molecules such as octreotide (OCT) provides selective interaction with the tumor cells. Also, a further functionalization with alginate (ALG) promotes increased lung accumulation compared to citrate-AgNPs and OCT-AgNPs [134]. The functionalization with homing peptides RPARPAR-OH (RPARPAR-AgNPs) demonstrates the preferential accumulation in the perivascular veins around pulmonary veins compared with the non-functionalized AgNPs. Accumulation in the lungs was nine-fold greater for functionalized than non-functionalized AgNPs [140].

Although very common in AgNPs synthesized using plant extracts as reducing agents, bacteria or other microorganisms (biosynthesis) is not exclusive to these AgNP formulations. Some works report very few physicochemical data or even do not report them at all, making it very difficult to analyze the influence of the different physicochemical properties on the reported toxicity. Yang, et al. found lung inflammation by quantifying the neutrophil increase in BALF with a size- and dose-response relationship on rats exposed to commercial AgNP-containing spray products after intensive and non-intensive applications. Using a compartmentalized physiologically based the alveolar deposition (PBAD) model to evaluate lung burden, the authors estimate the transfer from the interstitial region to lymph nodes as the immediate risk of AgNPs. However, the prediction has a minimal scope because neither reported the coating agent of AgNPS (if it exists) or the rat strain used for the evaluations [136]. Roda, et al. report lung parenchyma injury after intratracheal instillation of 50 μg/rat of 20 nm AgNPs obtained from a 1% water suspension. However, no more physicochemical details were provided regarding AgNPs’ formulation composition [137].

Other works even use AgNPs with different coatings indistinctly (citrate and PVP) to carry out the evaluations without distinguishing the effects that each one can produce. They only consider the similarity in size to identify lung damage after intratracheal instillation of 10 μg Ag/male ICR mouse [131]. These results make it even harder to identify the toxicological effect of each AgNP formulation, leading to a misleading conclusion that all AgNPs can be treated the same as discrete molecules. Sampath, et al. report the obtention of An-AgNPs and Py-AgNPs using the extract from Acacia nilotica (An) and pyrogallol (Py), respectively. The administration of An-AgNPs and Py-AgNPs to male Wistar rats by intraperitoneal injection at doses of 20 and 40 mg/bodyweight for 14 days once a day shows no damage to major target organs, liver, kidney, spleen, lungs, and heart [141]. He, et al. reported the antitumor activity of 8–22 nm AgNPs obtained from longan peel powder used as a reducing and stabilizing agent on human tumor lung H1299 cells xenografted on female severe combined immunodeficient (SCID) mice. A significant tumor volume decrease compared with the control group was observed after intraperitoneal administration of 10 μg/g of bodyweight for 36 days [135].

The preceding makes it evident that the design of new biocompatible nanomaterials must consider the type of coating and its adequate proportion to fulfill its therapeutic activity and provide a longer useful life that allows adequate waste treatment. A fundamental fact for all AgNP formulations evaluated, regardless of the coating reported, is that it does not exceed 5% of the composition of the formulation in any case. Although the cytotoxicity is reduced in the presence of the coating agent, having a minimum quantity only slows down the release of silver ions, which would explain the damage observed in prolonged exposure times and, in some cases, the subsequent recovery.

### 4.6. Human Exposure

The widespread use and highly demanded production urge the workplace investigations of occupational exposure to silver [91]. Silver accumulation will have a direct impact on toxicity through persistence over time. Several activities related to silver daily exposure have shown acute and chronic toxicity cases deriving in argyria, arterial blood oxygen decrease, heart rate increase, and even lung failure. Moreover, exposure to AgNPs is linked to significant recirculation times in the lungs compared to Ag ions [7] (Table 3).

Interestingly, there is evidence that the general population has a certain level of this metal in the blood (1 μg silver/L) even without silver occupational exposure. However, the worldwide use of AgNPs, which promotes an indirect human exposure to different forms of silver, will be a daily concern from now on, and it will only be enhanced. Hence, if we want to prevent the harmful effects of Ag from AgNPs or the release of Ag ions from AgNP dissolution, it is necessary to standardize our evaluation models. Notably, in vitro and in vivo pulmonary models to carry out physiologically relevant absorption, distribution, metabolism, and excretion (ADME) studies to develop safer AgNP formulations.

## 5. Lung Models as a Tool for Evaluating ADME to Predict Environmental Implications

ADME studies are widely used in pharmacology to evaluate the disposition within the organism of a drug of interest. As we have established before, AgNPs appear to be an excellent selective pulmonary chemotherapy; however, several implications must be considered before moving on to AgNP-based drug development. First, we must assume that airborne AgNPs will absorb within the lungs from inhalation exposure and may be able to trespass the air–blood barrier and epithelial cell barrier to disperse into remote organs [126]. It also should be considered that AgNPs could be metabolized by the lungs and the different organs to which they are biodistributed. Finally, this will imply an imminent excretion of the organism, which will end up in the water, soil, and air.

AgNPs can interact with organic matter (OM) and cationic species such as sodium and calcium, affecting their ionic strength and stability in aquatic environments. Interestingly, AgNPs have proven more stable under interaction with OM than while interacting with sodium and calcium [142]. These effects are further related to AgNPs’ coating agents interacting with OM and ionic species [143]. Destabilization of AgNPs will additionally result in Ag^+^ rapid release, resulting in imminent cytotoxicity for either microorganisms or humans [144]. Some other studies report that AgNPs can interact with humic substances (HS) present in the soil. Several authors state that soil properties such as pH directly impact nanomaterial agglomeration and functionalization [135,145,146,147]. The addition of H_2_S to water solution containing citrate-AgNPs increased pH value from five to eight and promoted the increase in AgNP size [145]. AgNPs under interaction with HS^−^ become more stable by forming a capping agent that provides both charge and steric stabilization [148].

The manufacture of silver vapors, fibers, and nanoparticles and their correlation with human exposure is widely reported [91]. The results of this accidental exposure indicate a higher level of bioaccumulation in the lung in both mice and humans (Table 2 and Table 3). The study subjects have shown high levels of silver in the blood, which are slowly eliminated through feces and urine. The elimination of AgNPs has been reported to be slower in the organism compared to the direct formulation of ions. Hadrup and coworkers attributed silver cytotoxicity to Ag^+^ liberation and their release rate [91]. Lung affectation at ultrastructural, metabolic, and genetic levels produced by Ag^+^ can be precisely measured by studying the differential expression of genes (DEGs).

## 6. Mechanisms of AgNPs Cytotoxicity on the Lungs

### 6.1. Differentially Expressed Genes (DEGs) Associated with Lung Cytotoxicity

The increasing use of nano drugs, their manufacture, and their release into the environment has made it difficult to predict the long-term damage to our health. Therefore, new tests to study the relationship between the nanomaterials and their target biological models have been developed. Inside the most precise assays to evaluate the PP x BM interaction, we have the differential expression of genes. Differentially expressed genes (DEGs) induced by AgNPs are a clear indicator of lung cell cytotoxicity [149]. Gene expression is commonly measured by DNA or mRNA quantification related to protein expression levels [150]. Multiple experimental procedures such as reverse transcription-polymerase chain reaction (RT-PCR), Elisa, Western blot, Northern Blot, Ribonuclease Protection Assay (RPA), mRNA differential display, SAGE, and DNA microarrays have been used to elucidate DEGs [151,152]. After AgNP exposure, the most frequent DEGs identified are DNA damage, ROS production, necrosis, apoptosis, cell membrane damage, mitochondrial disruption, autophagosome formation, lysosomal dysfunctions, epithelial-mesenchymal transition (EMT), and cellular senescence (Table 4). Among the most over-/under-expressed genes are matrix metallopeptidases, superoxidases, dehydrogenases, nuclear Factors, apoptotic proteins, heme oxygenases, tumor suppressor proteins, interleukins, and mucin regulators (Table 4).

In this sense, the differential expression of genes has been a great tool in studying new therapeutic targets of attacks on tumor cells. The upregulation of genes such as p53 and p21 represents tumor suppression; Bax, Bid, Cyt C, and Bak are associated with apoptotic and necrotic pathways. On the other hand, downregulation of Bcl-2 and Bcl-XL are characteristic of apoptotic and necrotic pathway blockers. The upregulation of specific genes shows that lung cancer cells are eliminated selectively (Table 4), but also gene upregulation in healthy cells was found, namely TGFβ1, MMP2, NOXO1, SOD2, sftpd, mir146, mir155, NOTCH3, MRAS. Gene expression patterns could indicate adverse health effects that can lead to DNA damage, ROS production, necrosis, apoptosis, cell membrane damage, mitochondrial disruption, autophagosome formation, lysosomal dysfunctions, cellular senescence, and epithelial-mesenchymal transition (EMT) indicative of carcinogenesis. These results highlight the importance of gene expression study in vitro and in vivo for a correct prediction of lung disease and progression.

Genes may signal different responses; thus, the in vitro model cannot be dissociated from gene expression. Genetic sequencing independent of the model show effects on ADN, cell arrest, and increased expression of pro-inflammatory genes such as IL-6, IL-8, mir146, mir155, mir21, and mir224 [59]. Several authors demonstrate proapoptotic genes increase (Bax, Casp3, Casp9, and miR-192), elevated levels of small airway epithelial repair and bronchiolar re-epithelialization (Duox1, Ect2, sftpa1, sftpd, muc1, and cftr), increase of epithelial-specific genes (MT1A and MT2A), and genes related to ROS overproduction (NOXO1 and SOD2). Besides, a decrease in the expression of anti-apoptotic gene Bcl-2, and overexpression of SLC26A (an important mucin promotor) were found [18,49,153]. Interestingly, it is possible to identify a coating-dependent gene expression from Table 4. Casp3 is upregulated by uncoated [59] or plant extract-AgNPs [52]; on the contrary, PVP-AgNPs downregulate [41] or do not modify [27] the casp3 expression. The same behavior was found for p53 expression. Wogonin-AgNPs formulation increases its expression [113]; meanwhile, PVP-AgNP produces no change [27] or downregulation [41] in p53 expression.

Hence, the correct study of the physicochemical properties of AgNPs and their relationship with DEGs in each lung model is of prime importance in determining AgNPs’ toxicity. The systematic knowledge of these parameters leads to specific modifications of physicochemical properties that improve the chemotherapeutic use of AgNPs, particularly for lung cancer treatment.

### 6.2. Trojan Horse Mechanism Exerting Lung Cell Death Mainly by p53 Apoptotic Pathway

Voluntary and involuntary exposure to nanoparticulate material is an everyday process. After inhalation, the increasing number of airborne nanoparticles reach our lungs, the first defense mechanism of our body against them. AgNPs can cross various physical and biochemical barriers, internalized by alveolar and bronchial cells, and some others will permeate through the gas exchange surface, accessing the systemic pathway, distributing throughout the body, and inducing chemotaxis. Even though very few studies have been carried out to elucidate the possible health effects that prolonged exposure to these external agents can bring about.

The Trojan horse mechanism is the process by which AgNPs are internalized to exert cytotoxicity within the cell. More precisely, AgNPs are internalized by lung cells and undergo dissolution, liberating high loads of toxic ions [154]. Once the AgNPs are internalized, released silver ions or AgNPs which are translocated directly to the mitochondria or the nucleus, promoting oxidative damage that could lead to cell death. The main routes of AgNPs cell internalization are endosomal or lysosomal endocytosis, diffusion, lipid peroxidation, and disruption of the membrane [155]. Even larger AgNPs, incapable of being endocytosed, have been demonstrated cytotoxic effects via receptor-mediated transduction pathways [156]. The first series of damaging effects at the cellular level will be observed at the membrane level, causing a decrease in its rigidity or rupture.

In the same way, ionic channels can be blocked physically by the agglomeration of AgNPs or functionally by Ag^+^ ions released from AgNPs dissolution. Once in the membrane, both processes prevent the correct cellular homeostasis and cell functioning. Finally, these species can interact with the protein receptors of the membrane, activating ROS signaling pathways [155,157]. These factors decrease the potential of the mitochondrial membrane and affect the structure of nuclear material interacting with the disulfide bridges of proteins and antioxidant molecules such as glutathione, generating structural damage to the genetic content and the deregulation of pathways and genes essential for cell survival [14,34]. The mitochondrial disruption and oxidation damage elicited by ROS overproduction was mainly attributed to the accumulation of silver ions; however, recent works demonstrate that the whole AgNPs could produce cellular injury with a negligible contribution of Ag^+^ release [158,159].

This review shows an exhaust investigation of lung high-value DEGs genes reported in Table 4. The differential gene expression observed among several lung cell types exposed to different AgNP formulations indicates that independently of the coating, size, or cell type, AgNPs impact transcriptional reprogramming of exposed cells. The specific gene expression changes of lung cells exposed to AgNPs can be used as inflammation or lung damage biological biomarkers [14,29,48,49,56,72,73,148,152]. Moreover, AgNPs exert cellular lung death by altering cell cycle, DNA content, and membrane tissue remodeling, further deriving cell death in a cell-type dependent manner. The p53 overexpression leads to apoptosis as the primary cell death pathway, and in some cases, necrosis and senescence could be observed (Figure 3).

## 7. Impact of Model Complexity in Lung Research

As the models increase in complexity, they show a better representation of the physical and biological barriers of the lungs. Co-cultures and 3D models provide relevant physiological information about significant lung barriers, e.g., air–blood barrier, pulmonary epithelial barrier, and airway barrier. The new variables (e.g., different lung cell types, macrophages, endothelial and epithelial cells, and extracellular matrixes) can provide additional measurable parameters in vitro corresponding to recent cellular interactions between two or more cell types. Cell–cell interactions and paracrine signaling by dissolution factors will govern these new communications. When there are two or more different lineages of cells, they will compete for the existing nutrients in the medium through the secretion of these factors, inhibiting the growth of their counterparts or, failing that, enhancing it [160].

Growth factor signaling is divided into three major modes: paracrine, autocrine, and juxtracrine. The autonomous production of the receptor, the ligand within the same cell system, and the union between them will occur between the same type of cells characterize the autocrine signaling. Otherwise, receptors and ligands are produced on different cells in paracrine signaling. Finally, juxtracrine signaling will occur only in tiny spaces through direct cell–cell contact since growth factors will be anchored to the membrane [160]. The effective selection of complex model components and the correct interpretation of the extracellular microenvironment response will be strongly related to the different signaling modes.

The paracrine and juxtracrine signaling of co-cultures and 3D cultures will promote resilience to the cytotoxicity elicited by AgNPs on the system. When AgNPs interact with heterogeneous groups of cells, an increase in the total number of metallothioneins and antioxidant proteins will occur. According to Table 4, genes such as MT1A, MT2A, SOD1, and SOD2 overexpress due to transcriptional reprogramming elicited by AgNP exposure [59,60]. The effects of these DEGs show how these complex systems promote defense mechanisms against the release of Ag^+^ and the ROS derived from this process. MT are proteins characterized by their binding to toxic metals through their chelation with cysteine residues, thus reducing the generation of reactive oxygen species [131]. On the other hand, SOD1 and SOD2 stand out for encoding antioxidant enzymes that, by binding to metal ions, can attach to ROS and degrade them to harmless products such as diatomic oxygen [161]. Co-cultures and 3D cultures were not significantly affected with similar concentrations of AgNPs compared to monocultures due to additional physical barriers present in more complex (Table 1) [29,49,81,82].

Intricacy level also resulted in a more specific prediction of biological endpoints indicative of lung disease. The differential expression of genes elucidated cellular response pathways characteristic of the cellular defense and repair mechanisms observed in vivo. This framework shows the relationship between the complex models reported in Table 1 and the observed outcomes in Table 2. Among these, we can find an increase in the number of neutrophils and macrophages characteristic of pulmonary inflammation and fibrosis, remodeling of lung tissue, the overproduction of mucosa, and the activation of the Nrf2 pathway characteristic of protection from the effects produced by ROS. It is essential to clarify that simpler models, such as monocultures, are unable to simulate a tissue’s cellular microenvironment, providing only the first approach to the toxicity profile.

Interestingly, biological endpoints turned out to be selective towards tumor cells at different levels of convolution. The effects are clear when concentrations up to 10 orders of magnitude lower for tumor cells are compared to those recorded for healthy cells [41,52]. In addition to the complexity level of the model, considerable dependence on the physicochemical properties could be noted. PVP-AgNPs produce less cytotoxicity even when internalized by healthy and cancerous cells (Table 1), attributable to the stability provided by the different coating agents that control the liberation rate of Ag^+^ ions in solution. The selective effect of AgNPs is allocated to the increased rate of internalization denominated enhanced permeability and retention effect (ERP), characteristic of tumor cells, and enhanced by the correct coating agent in the precise amount (Table 1) [162]. In lung complex models ruled by autocrine signaling, the tumor microenvironment makes these systems even more resistant to AgNPs than monocultures [27].

The increased selectivity of cell death induction in tumor cells by different ANP formulations compared to healthy models is through apoptotic processes, specifically the p53 pathway. For the evaluation of selective activity, it is possible to evaluate the overexpression of p53 for the BEAS-2B, A549 models and in primary lung models exposed to AgNPs [11,26,49,65,108]. In turn, traces of an effect were found in the overexpression of the proapoptotic genes P53, Cas3, Cas9, Bax, Bid, and Bak, and an underexpression in the apoptosis-blocking genes Bcl-2 and Bcl-xL for these cell models [26,49,65,108,141]. However, selectivity lies in the overexpression and underexpression of the genes for each model. In healthy models, a 1.8-fold increase in p53 expression compared to expression four times lower than found in the A549 line [11,108].

Independently of the route, the administration resulted in bioaccumulation in great quantity in the lungs, according to in vitro and in vivo toxic damage and biological endpoints indicatives of lung disease (Table 2). As a matter of fact, uncoated AgNPs bioaccumulate in a dose-dependent manner with significant frequencies in the lungs [105,107,149,150]. Uncoated AgNPs tend to form a protein corona easily; hence, bioaccumulated AgNPs generate lung cytotoxicity with a time-dependent partial recovery system. For the extrapolation of in vivo toxicity to humans (Table 3), it is imperative to say that the occupational exposure to silver in all ways corresponded with a rise in silver levels in the blood. The organism can overcome the acute and chronic toxicity derived from the exposure to silver in most cases through excretion. However, in vitro and in vivo toxicity results suggest greater toxicity of silver at the nanometric scale. These uncoated or unstable coated AgNPs have generally exhibited chronic and acute lung cell death in a dose-dependent manner. Therefore, the results urge us to stop the overproduction of unstable formulations of AgNPs before this can become an environmental problem and a potential risk to human health worldwide is merely needed. 

Even though the in vivo models can estimate some pathological lung changes, there still are differences in anatomy and physiology that allow a limited representation of the characteristics presented in humans. Several factors such as pathology biomarkers, acinar cell size, the extension of the airways, and even the thickness of the blood–air barrier between species contribute to a different response [153].

Using primary human lung cells incorporating different cell types provides greater physiological relevance; thus, co-culture models and 3D cultures demonstrate an efficient next step in obtaining relevant physiological information in lung models. The BECs, alveolar cells, macrophages, and pneumocytes, in an environment rich in pulmonary and mucosal surfactant at the air–liquid interface, could constitute a feasible model to evaluate lung toxicity, as shown in Figure 4. The complex arrangement allows a significant improvement in the simulation of the cellular microenvironment in the lungs and the dynamics of interaction with AgNPs. However, efforts must focus not only on the standardization of the in vitro models more representative of human models but also on the physicochemical properties that should be reported to identify the damage produced by each AgNP formulation.

## 8. Conclusions

The study of AgNPs’ interaction with lung cells shows selective toxic effects for the different cell models attributed to the other coating agents. In particular, the increase in the complexity of the model yielded a rise in the level of total measurable cytotoxicity. Mono-cultures, 2D, 3D, and ex vivo models were physiologically relevant for ADME studies, similar to in vivo models. Despite many groups still associating cytotoxic damage primarily with the release of Ag^+^, we have shown that AgNPs’ toxicity also depends on the size, coating agent, composition of the formulation, and the complexity of the cell model.

The uptake, biodistribution, and metabolism of the different formulations of AgNPs promote a different effect for each type of lung cell. These effects proved to be precisely measurable by the analysis of DEGs. The study of these results exhibited distinct biological endpoints indicative of damage progressions, such as pulmonary inflammation and fibrosis after AgNP exposure, regardless of the administration route. The nature and amount of coating agent present in the AgNP formulations strongly contribute to cytotoxic damage, biodistribution, and interaction with specific targets in the lung. In general, natural and synthetic polymeric coatings provide significant advantages in cytotoxic and biodistribution modulations compared with coating agents that provide less stable formulations, such as citrate, and an even better response than phytogenic or uncoated formulations.

This means that the physicochemical characterization of AgNPs must be as complete as possible and not be limited to indicating the size of the metallic nucleus and the possible functional groups present in the molecules that serve as coating. It is clear that AgNP production continuously increases due to their remarkable properties, thus, emphasizing the need to produce more stable formulations, avoiding the manufacture of uncoated nanoparticles since their high degree of toxicity and low selectivity. Even though many formulations may produce similar effects based on the biomarkers that we measure, it is imperative to identify that each formulation will generate specific damage based on its physicochemical properties and the type of cell with which it is interacting.

It is necessary to identify the amount of silver contained in each formulation, the type and amount of substance used as a coating agent, the surface charge, and the stability of the AgNPs formulation in the different culture media to establish its decomposition kinetics. Having information on these physicochemical properties will allow us to develop a baseline of potentially toxic effects of AgNPs, particularly in a complex system such as the lung.

With all this in mind, the direction of our future efforts is evident, systematizing the reported physicochemical data and the in vitro, ex vivo, and in vivo models used for toxicity evaluations. In addition, the development of new coatings and stabilization methods for future formulations of AgNPs will allow a longer useful life and, in turn, will provide additional time for adequate treatment of waste after use, reducing the critical environmental footprint generated. This will also contribute to establishing and strengthening specific regulations on health and the environment.

## Figures and Tables

**Figure 1 nanomaterials-12-02316-f001:**
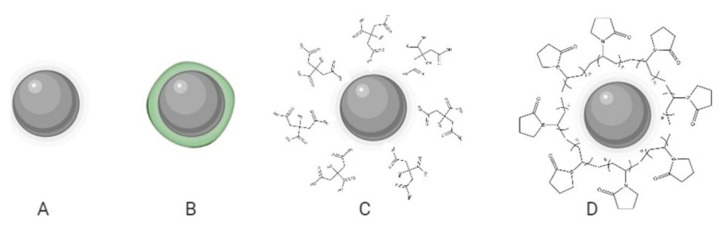
Schematic representation of AgNP formulations: uncoated (**A**) and with different coating agents; phytogenic (**B**), citrate (**C**), and PVP (**D**).

**Figure 2 nanomaterials-12-02316-f002:**
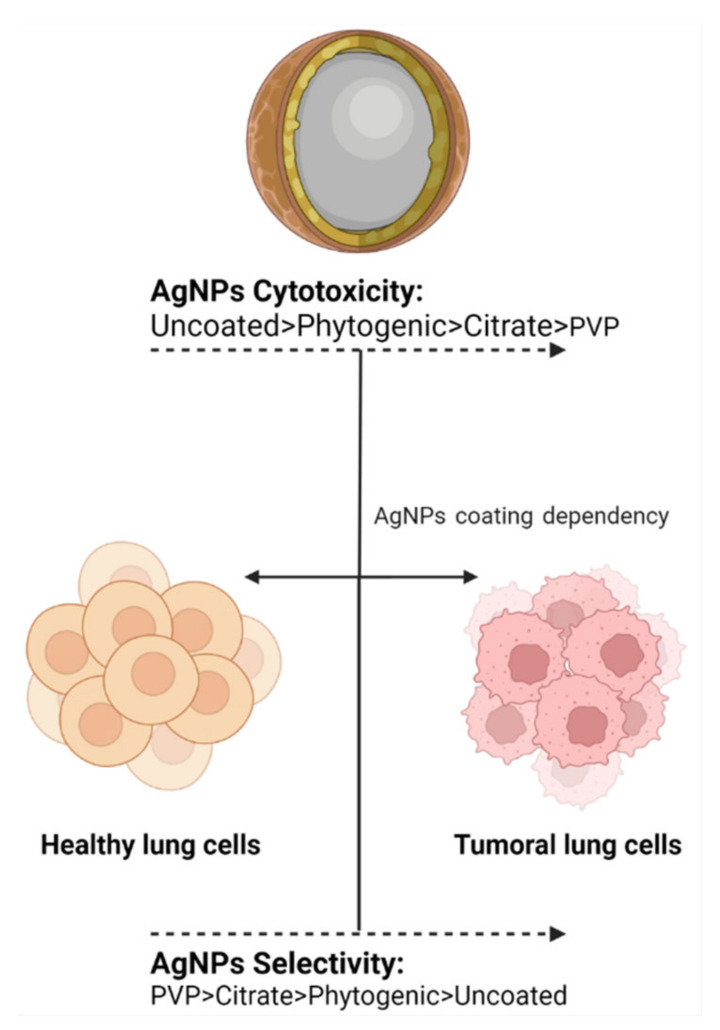
The coating agent helps in modulating cytotoxic activity and antiproliferative selectivity. The image shows the trend observed for the most frequently used coating agents in AgNPs.

**Figure 3 nanomaterials-12-02316-f003:**
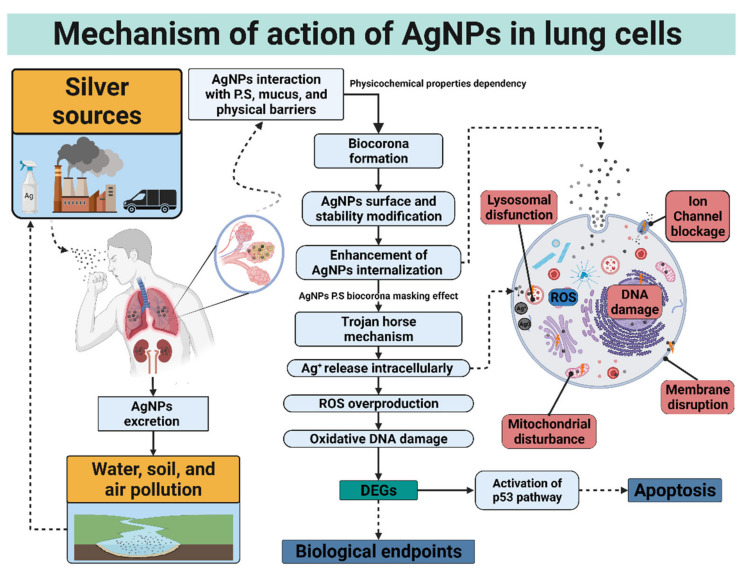
Continuous exposure to AgNPs or silver ions generates imminent bioaccumulation in major organs, specifically in the lungs. AgNPs will interact with the different biological components existing in the lungs, facilitating the formation of protein corona and biodistribution. Once inside, AgNPs exert a cytotoxic effect through a Trojan horse-type mechanism for uncoated or non-stable coating agents or a different mechanism when the coating agent is most stable or at a high concentration. It could generate ROS overproduction, transcriptional reprogramming, and apoptosis mediated mainly by the p53 pathway. This image summarizes the proposed cytotoxic mechanism of AgNPs within the lungs.

**Figure 4 nanomaterials-12-02316-f004:**
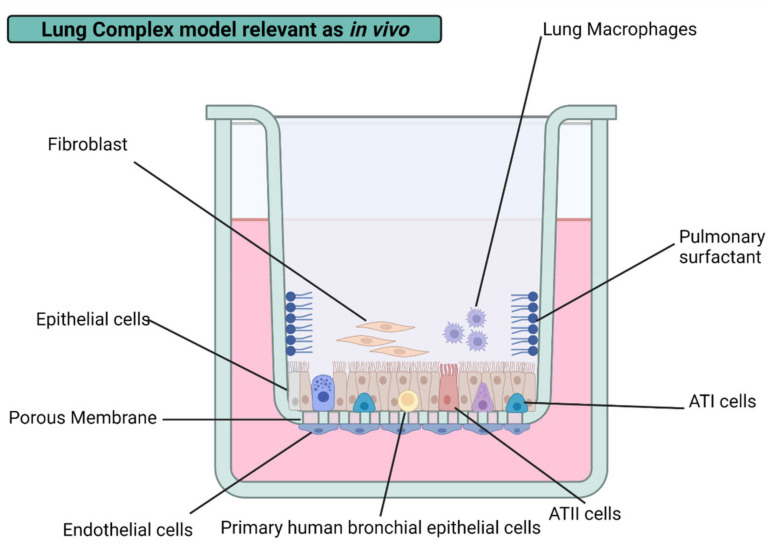
The image shows a proposal for a 3D model and the components that should be incorporated to generate better predictor models of the respiratory toxicity of AgNPs in vitro in future lung toxicology research. The addition of primary bronchial, epithelial, endothelial, fibroblast, ATI, and ATII cells will allow the in vitro representation of the physical barriers of the lungs. In turn, the addition of components such as pulmonary surfactant and macro-phages will allow the correct simulation of the cellular microenvironment and the immune defense system.

**Table 1 nanomaterials-12-02316-t001:** Cytotoxic effect of AgNPs in different lung cell lines.

Coating	Size (nm)	Concentration (μg/mL)	IC_50_ (μg/mL) ^a^	Exposure Time (h)	Cell Line	Outcomes	Cytotoxic Response	Ref
Monocultures								
PVP	20	0, 10, 25, 50, 100 and 200	100	24, 48, 72	A549	Gene and protein expression decreases of p53, p21, MDM,2, and caspase 3. Mitochondrial ROS production. Global acetylation levels decrease on tails of histone H3 protein. Global DNA methylation increases. Late apoptosis/necrosis increase after 48 h. HMOX1 has a high expression on A5 and 49, might it render them less susceptible to ROS-induced cell death early-stage apoptosis.	Concentration-, and Time-dependency.	[41]
PVP	10, 20	5–10	10 nm: 56.4 20 nm: >100	24, 48	A549	Severe ADN damage. Cell cycle arrest, increase in several of cells at S and sub-G1 phases (DNA repair mechanism more effective on 10 nm AgNPs). A decrease in cell viability. Increase of late-apoptotic and necrotic cells at 100 μg/mL.	Size-, Concentration-, and Time-dependency.	[42]
PVP	23	1–10	NS	24, 48, 72	A549 Calu-1 BEAS-2B NCI-H358	Cell cycle arrest. Cell viability decreased in all cell lines except NCI-H358. Mitochondrial ROS production and protein oxidation, particularly on AgNPs sensitive cell lines. Decrease in cellular ATP levels. Cell arrest on G2 and S-phase for A549 and Calu-1 and S-phase for BEAS-2B. NCI-H358 cells did not show cell cycle changes related to AgNPs exposure.	Concentration-, Time-, and Cell type dependency.	[43]
PVP	25	0.4, 1, 4, 10	>100	240	NHLF MRC-5	Moderate acute toxicity for MRC-5 and cellular senescence using sub-toxic concentrations associated with β-galactosidase (SA-β-gal) activity and heterochromatin foci (SAHF) Expression of SASP and inflammatory genes G2/M phase arrest completed after 10 days 685 transcripts upregulated and 718 transcripts downregulated in RNA-seq global mRNA levels Potential role of the COX2-PGE2 pathway in AgNPs-induced lung cellular senescence. COX2-PGE2 pathway regulated by p65 and highly differentiated. BCL-2 downregulated by AgNPs subsequently undergoes apoptosis.	Concentration-, and Cell type-dependent.	[44]
PVP	50 and 200	5.6, 11.5, 22.5, 45	NR	16	NR8383	Increase of lactate deshydrogenase (LDH) and glucuronidase (GLU) activity. TNH-α increase at lower concentration of 50 nm citrate-AgNP and at the higher concentration of PVP-AgNP	Concentration-dependent	[40]
Shikonin	20	0.078–10	2.4 ± 0.11	24	A549	Cell viability and proliferation decrease.	Concentration-, and	[45]
*Acacia nilotica*, NG, or TKP	10–78	10–100	Wi38: 86.15 A549: 65.85	12, 24, 48	A549 Wi38	Cytotoxic selective to cancer cells. Inhibition of cell cycle. ROS mediated apoptosis.	Cell-type-dependent.	[46]
Gallic acid	10–30	5, 25, 50, 100, 200	46.5	24	A549	Effective in treating the radiation toxicity and resistance developed by the cancer cells during cancer treatment. Cell viability decrease. Epithelial-Mesenchymal Transition suppression.	Concentration-dependent.	[47]
*Caulerpa taxifolia*	10–100	10–100	40,000	24	A549	Morphological damage and condensation morphology. Cell death. Apoptosis/necrosis induction.	Concentration-dependent.	[48]
*Avicennia marina*	10–20	10–80	50,000	24	A549	Cancer cell growth inhibition. Damage to the mitochondrial membrane. ROS.	Concentration-dependent.	[49]
*Tinospora cordifolia*	25–50	25, 50, 75, 100, 150	100	12, 24, and 48	A549	Cell viability decrease. Cytomorphological changes. Apoptosis. Nuclear damage. ROS. Loss of mitochondrial membrane potential (ψm).	Concentration-, and Time-dependent.	[50]
*Wogonin*	5, 40	2–10 μM 1–5 μM (Ag content)	5 nm: 2 μM 40 nm: 6 μM	24 and 48	A549	Cell viability decrease. ROS. Activation of the mitochondrial apoptotic pathway. DNA damage. Activation of Caspase-9 and Caspase-3. Secretion of pro-inflammatory markers such as TNFα.	Concentration-, and size-dependent.	[51]
*Artemisia oliveriana*	10.63	5, 25, 50, 100 and 200	A549: 3.6 MRC-5: 10	24	A549 MRC-5	Cell viability decrease. Apoptotic genes Bax, Casp3, Casp9, and miR-192 expression increase. Anti-apoptotic gene Bcl-2 expression decrease. Cell cycle shift to sub-G1 phase. Antioxidant activity. Fewer effects on normal cells (MRC-5). Fragmentation of the genomic DNA.	Concentration-, and Cell-Type dependent.	[52]
*Toxicodendron vernicifluum*	2–40	5, 10, 20, 40, 80, 160, 320	A549: >320 NiH3T3: 160	24	A549 NIH3T3	Cell viability decreased on A549 but not on mouse embryo cells. ROS mediated apoptosis on A549. 95% Cell death at the maximum concentration for A549.	Concentration-, and Cell type-dependent.	[53]
Citrate	10, 75	1	Not specified	144	BEAS-2B	719 down-regulated and 998 up-regulated genes after exposure. DNA damage, Cell cycle arrest on G1. Fibrosis induction. EMT (epithelial-mesenchymal transition). Cell transformation is indicative of an oncogenic phenotype.	Concentration-, Size-dependent-, and Time-dependent.	[14]
Citrate	60	50, 100, 200	200 μg Ag/mL	24	A549 HPSAEpiC	Lysosomal pH alkalization (dysfunction) and autophagosome formation. Inhibition of autophagic flux. Inhibition of Transcriptional Factor EB (TFEB) expression. Concentration-dependence increase of p62 and LC3B-II proteins.	Concentration-, And Cell type-dependent.	[54]
Citrate, chitosan	7–10	6.25 × 10^12^, 1.25 × 10^12^, 2.5 × 10^12^, 5 × 10^12^ NPs/mL	NHBE: 0.7 μg/cm^2^ A549 and BEAS-2B: not in range	0.5, 4, and 24 hours	A549. BEAS-2B. NHBE.	No cytotoxicity was observed on A549 and NHBE; not responsive to Transepithelial/transendothelial electrical resistance (TEER) change. Higher cytotoxicity resistance for NHBE compared with the other cells. ROS production is most prominent in A549.	Concentration-, Cell type-, and Coating dependent	[55]
Citrate	10, 75	2 and 10	10	24 and 48	HLF-1	Decrease in cell viability. Reduction of metabolic activity. Procollagen and proinflammatory cytokine secretion.	Time-dependent-, Concentration-, and size-dependent	[28]
Uncoated	4.7, 42	0.84–2000	4.7 nm: 7700 42 nm: 1150,000	24	HbPF	Decrease in HPF viability. Reduction in cell mitochondrial activity and LDH leakage. ROS production and oxidative stress. No statistically significant changes in SOD activity. GSH depletion.	Size-dependent.	[56]
**Co-cultures**								
Starch	20 ± 4	7.25 μg, 41.25 μg (Nebulization)	Out of range	24	hAELVi and THP-1	High viability. Problems with determination.	Concentration-, and Cell type-dependent.	[57]
Garcinia mangostana	12	2.5 μg/mL	Out of range	24	A549 with BEAS-2B	Cell viability decreased for A549. BEAS-2B is highly resistant.	Cell type-dependent.	
Tannic acid	50 ± 4	3 mg/L, 30 mg/L	Out of range	24	Calu-3, EA.hy926, and THP-1	High toxicity at high concentration treatment. Pro-inflammatory markers IL-6, IL-8, and TNF-α significant secretion reduction.	Cell type-, and Concentration-dependent.	[58]
**3D-cultures**								
Uncoated	14	1.5, 4.4 and 13.2 ng/cm^2^.	LDH (not specified)	6 and 24	Organotypic-reconstituted 3D human primary small airway epithelial cell	Neutrophil accumulation. Macrophage levels modestly increased. SLC26A4 mucin gene production overexpressed. Duox1 expression increased (Small airway epithelial repair and bronchiolar re-epithelialization). Ect2, sftpa1, sftpd, muc1, and cftr epithelial-specific genes increase. MT1A and MT2A were upregulated (Cellular defense systems are in place to mitigate the effects of metal ion exposure), and metal overload. mir146, mir155, mir21 and mir224 (inflammatory process). NOXO1 and SOD2 ROS, mitochondrial disruption, DNA damage, cell cycle regulation, G2/M phase cell cycle arrest. The inflammatory process, Immunomodulatory response, and tissue remodeling.	Concentration-dependent.	[59]
Uncoated	20, 200	0.05, 0.5, 5 μg/cm^2^	Out of range	6 and 24	3D model representative of the alveolar barrier	ROS, cell death Increased level of mRNA Antioxidant and anti-inflammatory HMOX-1. Nuclear translocation of the transcription factor NF-kB in endothelial cells. Inflammation, increase in the mRNA levels of IL-6 and IL-8.	Concentration-, and Size-dependent.	[60]
PVP	10–20	40	Out of range	24	3D and 2D A549 model	Apoptosis/Necrosis No effects on p53, Bax, and Caspase-3. Slightly reduced expression of Bcl-xL and NF-kB genes. Cells within 3D cultures were less affected by nanomaterials than in 2D cell cultures. Less affected when combined with hydra protein (ROS entrapment).	Concentration-, size-, and Model-dependent.	[27]

A549: human lung carcinoma (epithelial); BEAS-2B: human bronchial epithelium (normal); Calu-1: human lung epidermoid carcinoma (non-small-cell lung cancer); Calu-3: human lung adenocarcinoma (bronchial epithelial cells); EA.hy926: endothelial cells from the human umbilical vein; hAELVi: human alveolar epithelial cells; HLF-1: human lung fibroblast; HPF: primary cultures of pulmonary human fibroblasts; HPSAEpiC: human small airway epithelial cells; MRC-5: human fetal lung (male, normal); NCI-H 358: bronchoalveolar carcinoma (non-small-cell lung cancer); NG: natural gum; NHBE: normal human bronchial epithelial cells; NHLF: normal human lung fibroblast; NIH3T3: mouse Swiss NIH embryo (fibroblast); NS: not specified; THP-1: human acute monocytic leukemia; TKP: tamarind kernel powder; Wi38: human fetal lung (female, normal).

**Table 3 nanomaterials-12-02316-t003:** Occupational exposure to different forms of silver.

Ways of Exposure	Average Blood Levels	Study Population (Individuals)
Population in general that does not work with silver	1 μg Ag/L	26
Silver material manufacturers	0.00035 and 0.00135 mg Ag/m^3^, blood levels of 0.34 and 0.30 μg Ag/L	2
Recovery of silver from x-rays and photographic films.	0.085 and 1 mg Ag/m3, 0.03 y 0.17 mg Ag/m^3^ resulting in blood levels of 49 y 79 μg Ag/L, respectively.	2
Exposed to silver oxides and silver nitrates	Media: 19.5 μg/L; range: 11–84 μg/L	30
Silver powder manufacturing	Media: 10 μg/L; range: 0.5–62 μg/L	25
Recovery of silver in waste	Media: 10 μg/L	21
Scrap silver recovery, coin silver refinery, jewelry production	Media: 10 μg/L; range: 0.1–23 μg/L	98
Smelting, refining, and manufacturing of silver salts	Media: 11 μg/L	37
Exposed to silver aerosol	154.4 μg/L	1

**Table 4 nanomaterials-12-02316-t004:** Differential expression of genes in lung cell models.

Differentially Expressed Genes (DEGs)	Cellular Response Pathways	AgNP Size (nm)	AgNP Coating	Cell Line	Ref
p53 ↓, p21↓, Mdm2↓, caspase-3↓	Cell damage DNA damage Apoptosis	20	PVP	A549	[41]
ATM protein ↑, Heme oxygenase-1↑	Cell cycle DNA damage Apoptosis	10, 20	PVP	A549	[42]
Bax↑, Casp3↑, Casp9↑, miR-192↑, Bcl-2↓	Cell cycle Apoptosis	10.63	*Artemisia* *oliveriana*	A549 MRC-5	[52]
685 transcripts upregulated and 718 transcripts downregulated in RNA-seq global mRNA levels Bcl-2↓	Cell growth (Senescence) Cell Cycle	25	PVP	Normal human lung fibroblast (NHLF) MRC-5	[130]
p53↑, p21↑, Bid↑, Bax↑, Bak ↑, Cyt C↑, Bcl-2↓, Bcl-xL↓	Enriched signaling pathways; MAPK2, TNF, IL17, P13k-AKT, NF- Kappa B, Apoptosis	5, 40	*Wogonin*	A549	[113]
719 genes were down-regulated, and 998 genes were up-regulated. Collagen related (COL1A1, COL1A2, COL6A2, COL11A1, COL16A1, COL18A1, COL21A1) ↑ COL17A1 ↓, MMP2 matrix-metallopeptidase involved in the degradation of collagen (IV, V, VII, X) ↑ MMP11, and MMP19 inhibition of metalloproteases ↑ TGFβ1, an important pro-fibrotic growth factor and a key regulator of lung fibrosis and its receptor TGFBR1 ↑, BAMBI ↓, AGTR1, PGF, and PDGF ↑, CDH1 ↑, CDH 12 ↓, NOTCH3 ↑, MMP2 ↑, MRas ↑, HIF1α ↓, Antioxidant enzymes such as glutathione-S-transferases (GSTM1, GSTM2, GSTM3, GSTT2/GSTT2B) ↑, M NQO1 ↑, EPHX1 ↑, CAT ↓	Enriched pathways related to: Carcinogenesis, Hepatic fibrosis, ROS, regulation of epithelial mesenchymal transition.	10, 75	Citrate	BEAS-2B	[14]
TFEB↓, LC3B-II↑, LAMP1→, P62↑, C, Bax↑, Bcl-xL↓, C, Casp3→, NF-kB↓, p53→	Apoptosis Necrosis	10, 20	PVP	3D and 2D A549 model with and without hydra protein	[27]
493 differentially regulated transcripts SLC26A4↑, Duox1↑, Ect2↑, sftpa1↑, sftpd↑, muc1↑, cftr↑, MTA1 and MTA2 ↑, NOXO1 and SOD2↑, mir146, mir155, mir21 and mir 224↑	Nrf2 Regulation of inflammatory processes Regulation of metals DNA damage cell cycle regulation Inflammatory process Immunomodulatory response ROS Tissue remodeling Metal overload.	14	Uncoated	Organotypic-reconstituted 3D human primary small airway epithelial cell	[59]
HMOX-1↑, NQO1↓, SOD1↑ MT-1A, MT-1B and MT-2A↑, Casp7 ↑, FAS↑, HSP70↑, GST↓, VCAM1↑, ICAM-1↓, NF-kB↓, IL-6↑, COX-2↓, N	Nrf2 regulates inflammatory processes Metal binding antioxidant metallothionein Apoptosis	20, 200	Uncoated	3D model representative of the alveolar barrier	[60]
Drp1↑, p-Drp1↑, Opa1↓, Mfn2↓, Casp3↑	Fission Fusion Apoptotic	10–20	Uncoated	Sprague Dawley Rats	[138]

p53: cellular division and cellular destruction (tumor suppression); Cas-3: cell death process (apoptosis, necrosis and inflammation); Bax: anti-apoptotic regulator; BCL2: blocks apoptotic death; MMP2 (matrix metalloprotein ase-2): matrix-metallopeptidase involved in degradation of collagens (IV, V, VII, X); NQO1 (NAD(P)H dehydrogenase [quinone] 1): decodes for inducible multifunctional antioxidant enzymes; SOD1 (Superoxide dismutase 1): protein coding gene; NF-KB1 (nuclear Factor Kappa B subunit 1): DNA transcription and immune cell development; PTGS2/COX2 (prostaglandin-endoperoxide synthase 2): mediator of physiological stresses responses such as infection and inflammation; HMOX1 (heme oxygenase 1): inflammatory process and Fe homeostasis; IL6 (interleukin 6): cellular proliferation and long-term survival; Slc26a4: epithelial expressed/mucin regulation; NOTCH3: blood vessels maintenance; TGFβ1: cell growth, cell proliferation, cell differentiation, and apoptosis; sftpd: lung defense against foreign agents and toxins.

## Data Availability

Not applicable.

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
