# Peer review of "Lung Models to Evaluate Silver Nanoparticles’ Toxicity and Their Impact on Human Health"

_nanomaterials, 2022, doi:10.3390/nano12132316_

Round 1
Reviewer 1 Report
In this review, the authors discussed the toxic effects of silver nanoparticles in lung with the particular focus on different lung models. Overall, the review is interesting and comprehensive while a few comments need to be addressed.
First of all, the title of the review is very broad and general. There is only a small section covering the environmental implications. The authors could also consider including the key word “lung models” in the title.
section 6 Differentially expressed genes (DEGs) associated with lung cytotoxicity. This section can be incorporated into section 7 Mechanisms of AgNPs cytotoxicity on the lungs.
Could the authors comment on the lung microbiome and their potential role in lung inflammation and the responses to silver nanoparticles or any other type of nanomaterials?
Author Response
Reviewer comments
- First of all, the title of the review is very broad and general. There is only a small section covering the environmental implications. The authors could also consider including the keyword “lung models” in the title.
Thank you very much for your comment, we agree that the initial title was very broad. Considering the recommendations, we modify the title as follows:
Lung models to evaluate silver nanoparticles toxicity and their impact on human health
- section 6 Differentially expressed genes (DEGs) associated with lung cytotoxicity. This section can be incorporated into section 7 Mechanisms of AgNPs cytotoxicity on the lungs.
We appreciate your suggestion; we merge both sections to enhance the manuscript fluency.
- Could the authors comment on the lung microbiome and their potential role in lung inflammation and the responses to silver nanoparticles or any other type of nanomaterials?
We performed an exhaustive search for AgNP-lung microbiome interactions, but only a few results get. The information from those articles was included in section 2.5 Surface functionalization (lines 213-229)
Surface functionalization also favors AgNPs interaction with the lung microbiome [40]. Depending on their coating, AgNPs could alter lung microbiome after instillation, being more pronounced the effect with citrate-AgNPs compared with PVP-AgNPs. The latter significantly reduce the inflammation produced by ovalbumin in BLAB/C mice and produce no adverse effect on non-sensitized mice. On these mice lung microbiome was altered by AgNPs increasing the abundance of Actinobacteria, Bacteriodetes, Firmicutes and Proteobacteria [39]. On the other hand, AgNPs can interact with the lipopolysaccharides of the microbial wall present in Gram-negative bacteria through hydrogen bonds and electrostatic interactions. After exerting their antimicrobial power, AgNPs could generate a new coating with the lipopolysaccharides (AgNPs-LPS). The new coating favors an immune response through the interaction of LPS with the toll-like receptor 4 (TLR4) present to a greater extent in pulmonary macrophages [40]. This will generate its activation and therefore the production of cytotoxic inflammatory mediators such as cytokines, chemokines, and tumor necrosis factor-alpha (TNF α) which can exert lung adverse effects and even cancer in an indirect manner [41,42]. AgNPs-lung microbiome interaction is a very scarce topic for which further careful study is highly recommended. In this regard, the choice of AgNPs coating has shown to be so important that if it is wrongly chosen it can lead to serious consequences in the lungs and the environment, even due to causes unrelated to the AgNPs-cell interaction.

Reviewer 2 Report
A review entitled "Silver Nanoparticles in Lung Toxicity. The implication in Environmental and Human Health.", contains generalized results on the study of the possible effect on human lungs of silver nanoparticles of various origins, sizes, shapes and other differing parameters. The work is of scientific interest because, as the authors noted, there are conflicting data on the cytotoxicity of silver nanoparticles, which makes it difficult to develop rules for their use. The review contains extensive information based on data from studies of various human lung models. The text is written in good language, easy to perceive, the narrative is logical and consistent. The article is recommended for publication after minor changes.
1) Page numbering is broken. After page 1, the numbering starts again from the beginning, then this is repeated twice.
2) After line 312 there is too much free space not occupied by the text.
3) After line 762 there is too much free space not occupied by the text.
4) After line 910 there is too much free space not occupied by the text.
5) Line 1081. It is advisable to move the list of references to a separate page.
6) Lines 53-54. It is recommended to list the various applications of AgNPs, with links to sources.
7) Lines 230-231. It is recommended to add a link to the article https://doi.org/10.3390/mi12121480 , dedicated, among other things, to phytogenic AgNPs.
8) In Table 1, it is recommended to increase the width of the "Outcomes" column to improve perception.
9) Line 313 is an extra line.
10) In Tables 2 and 4, the interpretation of the terms is given in too small font at the end.
Author Response
We want to thank reviewer 2 for the careful lecture on our manuscript and the helpful feedback.
Reviewer comment:
- Page numbering is broken. After page 1, the numbering starts again from the beginning, then this is repeated twice.
- After line 312 there is too much free space not occupied by the text.
- After line 762 there is too much free space not occupied by the text.
- After line 910 there is too much free space not occupied by the text.
Thanks for your observations, all suggested modifications in reviewer comments 1-4 were done.
- Line 1081. It is advisable to move the list of references to a separate page.
Done
- Lines 53-54. It is recommended to list the various applications of AgNPs, with links to sources.
Thanks, we include three review articles to attend the comment.
These have found a wide range of applications in various areas such as textiles, agriculture, renewable energy, food, catalysis, bioremediation, and biomedicine [4–6]
- Abdelghany, T.M.; Al-Rajhi, A.M.H.; Al Abboud, M.A.; Alawlaqi, M.M.; Ganash Magdah, A.; Helmy, E.A.M.; Mabrouk, A.S. Recent Advances in Green Synthesis of Silver Nanoparticles and Their Applications: About Future Directions. A Review. Bionanoscience 2018, 8, 5–16, doi:10.1007/s12668-017-0413-3.
- Nicolae-maranciuc, A.; Chicea, D.; Chicea, L.M. Ag Nanoparticles for Biomedical Applications — Synthesis and Characterization — A Review. 2022.
- Garg, D.; Sarkar, A.; Chand, P.; Bansal, P.; Gola, D.; Sharma, S.; Khantwal, S.; Surabhi; Mehrotra, R.; Chauhan, N.; et al. Synthesis of silver nanoparticles utilizing various biological systems: mechanisms and applications—a review. Prog. Biomater. 2020, 9, 81–95, doi:10.1007/s40204-020-00135-2.
- Lines 230-231. It is recommended to add a link to the article https://doi.org/10.3390/mi12121480 , dedicated, among other things, to phytogenic AgNPs.
Thanks for the contribution, we include the article in the main manuscript line 252.
- In Table 1, it is recommended to increase the width of the ¨outcomes¨ column to improve perception.
We appreciate the recommendation. The table was modified by increasing the width of the outcome column.
- Line 313 is an extra line.
This extra line was removed from the main text.
- In Tables 2 and 4, the interpretation of the terms is given in too small a font at the end.
Thank you very much for the observation, we erroneously include the table footnote as a superscript. The footnote font size was corrected.
